# DEEP LEARNING MEETS NONPARAMETRIC REGRESSION: ARE WEIGHT DECAYED DNNS LOCALLY ADAPTIVE?

**Kaiqi Zhang**
Department of Electrical and Computer Engineering
University of California, Santa Barbara
*kzhang70@ucsb.edu*

**Yu-Xiang Wang**
Department of Computer Science
University of California, Santa Barbara
*yuxiangw@cs.ucsb.edu*

## ABSTRACT

We study the theory of neural network (NN) from the lens of classical nonparametric regression problems with a focus on NN's ability to *adaptively* estimate functions with *heterogeneous smoothness* — a property of functions in Besov or Bounded Variation (BV) classes. Existing work on this problem requires tuning the NN architecture based on the function spaces and sample size. We consider a "Parallel NN" variant of deep ReLU networks and show that the standard $\ell_2$ regularization is equivalent to promoting the $\ell_p$-sparsity ($0 < p < 1$) in the coefficient vector of an end-to-end learned function bases, i.e., a dictionary. Using this equivalence, we further establish that by tuning only the regularization factor, such parallel NN achieves an estimation error arbitrarily close to the minimax rates for both the Besov and BV classes. Notably, it gets exponentially closer to minimax optimal as the NN gets deeper. Our research sheds new lights on why depth matters and how NNs are more powerful than kernel methods.

## 1 INTRODUCTION

*Why* do deep neural networks (DNNs) *work* better? They are universal function approximators (Cybenko, 1989), but so are splines and kernels. They learn data-driven representations, but so are the shallower and linear counterparts such as matrix factorization. The theoretical understanding on why DNNs are superior to these classical alternatives is surprisingly limited.

In this paper, we study DNNs in nonparametric regression problems — a classical branch of statistical theory and methods with more than half a century of associated literatures (Nadaraya, 1964; De Boor et al., 1978; Wahba, 1990; Donoho et al., 1998; Mallat, 1999; Scholkopf & Smola, 2001; Rasmussen & Williams, 2006). Nonparametric regression addresses the fundamental problem:

- Let $y_i = f(x_i) + $ Noise for $i = 1, ..., n$. How can we estimate a function $f$ using data points $(x_1, y_1), ..., (x_n, y_n)$ in conjunction with the knowledge that $f$ belongs to a function class $\mathcal{F}$?

Function class $\mathcal{F}$ typically imposes only weak regularity assumptions such as smoothness, which makes nonparametric regression widely applicable to real-life applications under weak assumptions.

**Local adaptivity.** We say a nonparametric regression technique is *locally adaptive* if it can cater to local differences in smoothness, hence allowing more accurate estimation of functions with varying smoothness and abrupt changes. A subset of nonparametric regression techniques were shown to have the property of *local adaptivity* (Mammen & van de Geer, 1997) in both theory and practice. These include wavelet smoothing (Donoho et al., 1998), locally adaptive regression splines (LARS, Mammen & van de Geer, 1997), trend filtering (Tibshirani, 2014; Wang et al., 2014) and adaptive local polynomials (Baby & Wang, 2019; 2020). In light of such a distinction, it is natural to consider the following question: *Are NNs* locally adaptive*, i.e., optimal in learning functions with heterogeneous smoothness?*

This is a timely question to ask, partly because the bulk of recent theory of NN leverages its asymptotic Reproducing Kernel Hilbert Space (RKHS) in the overparameterized regime (Jacot et al., 2018; Belkin et al., 2018; Arora et al., 2019). RKHS-based approaches, e.g., kernel ridge regression with

Table 1: Comparison with the results in the literature

|  | # layers | Activation | Function space | Minimax rate | Remark |
|---|---|---|---|---|---|
| Parhi & Nowak (2021a;c) | 2 | truncated power | $BV^m$ | Yes | Non-standard activation and regularization (when $m > 1$). |
| Schmidt-Hieber (2020) | $\geq 3$ | ReLU | Hölder | Up to a log factor | With sparsity constraint. |
| Suzuki (2018) | $\geq 3$ | ReLU | Besov & m-Besov | Up to a log factor | With sparsity constraint. |
| **Ours** | $\geq 3$ | ReLU | Besov & BV | Up to $n^{o(1)}$ factor | Requires only $\ell_2$ regularization. |

any fixed kernels are *suboptimal* in estimating functions with heterogeneous smoothness (Donoho et al., 1990). Therefore, existing deep learning theory based on RKHS does not satisfactorily explain the advantages of neural networks over kernel methods.

We build upon the recent work of Suzuki (2018) and Parhi & Nowak (2021a) who provided encouraging first answers to the question above. Specifically, Parhi & Nowak (2021a, Theorem 8) showed that a two-layer *truncated power function* activated neural network with a non-standard regularization is equivalent to the LARS. This connection implies that such NNs achieve the minimax rate for the (high order) bounded variation (BV) classes. A detailed discussion is provided in Section B. Suzuki (2018) showed that multilayer ReLU DNNs can achieve minimax rate for the Besov class, but requires the artificially imposed sparsity-level of the DNN weights to be calibrated according to parameters of the Besov class, thus is quite difficult to implement in practice. (Oono & Suzuki, 2019; Liu et al., 2021) replaced the sparse neural network with Resnet-style CNN and achieved the same rate, but they similarly require carefully choosing the number of parameters for *each* nonparametric class. We show that $\ell_2$ regularization suffices for *mildly overparameterized* DNNs to achieve the optimal "local adaptive" rates for *many* nonparametric classes at the same time.

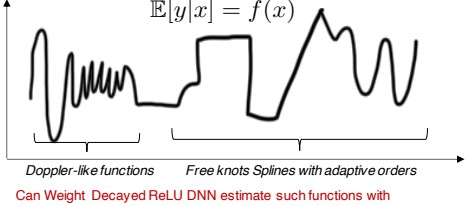

Figure 1: Illustration of a function with heterogeneous smoothness and the problem of locally adaptive nonparametric regression.

**Parallel neural networks.** We restrict our attention on a special network architecture called *parallel neural network* (Haeffele & Vidal, 2017; Ergen & Pilanci, 2021c) which learns an ensemble of subnetworks — each being a multilayer ReLU DNNs. Parallel NNs have been shown to be more well-behaved both theoretically (Haeffele & Vidal, 2017; Zhang et al., 2019; Ergen & Pilanci, 2021b;c;d) and empirically (Zagoruyko & Komodakis, 2016; Veit et al., 2016). On the other hand, many successful NN architectures such as SqueezeNet, ResNext and Inception (see (Ergen & Pilanci, 2021c) and the references therein) use the idea similar to a parallel NN.

**Weight decay,** also known as square $\ell_2$ **regularization**, is one of the most popular regularization techniques for preventing overfitting in DNNs. It is called "weight decay" because each iteration of the gradient descent (or SGD) shrinks the parameter towards 0 multiplicatively. Many tricks in deep learning, including early stopping (Yao et al., 2007), quantization (Hubara et al., 2016), and dropout (Wager et al., 2013) behaves like $\ell_2$ regularization. Thus even though we focus on the exact minimizer of the regularized objective, it may explain the behavior of SGD in practice.

**Summary of results.** Our main contributions are:

1. We prove that the (standard) $\ell_2$ regularization in training an $L$-layer *parallel* ReLU-activated neural network is equivalent to a sparse $\ell_p$ penalty term (where $p = 2/L$) on the linear coefficients of a learned representation (Proposition 4).

2. We show that the estimation error of $\ell_2$ regularized parallel NN can be close to the minimax rate for estimating functions in Besov space. Notably, the method can adapt to different smoothness parameter, which is not the case for many other methods.

3. We find that deeper models achieve closer to the optimal error rate. This result helps explain why deep neural networks can achieve better performance than shallow ones empirically.

Besides, we have the following technical contributions which could be of separate interest:

- We provide a way to bound the complexity of an overparameterized neural network. Specifically, we bound the metric entropy of a parallel neural network in Theorem 5, and the bound does not depend on the number of subnetworks.
- We propose a method to handle unconstrained function subspace when bounding the estimation error as in Equation (4).

The above results separate parallel NNs with any linear methods such as kernel ridge regression. To the best of our knowledge, we are the first to demonstrate that standard techniques ($\ell_2$ regularization and ReLU activation) suffice for DNNs in achieving the optimal rates for estimating BV and Besov functions. The comparison with previous works is shown in Table 1. More discussion about related works are shown in Section A.

## 2 PRELIMINARY

### 2.1 NOTATION AND PROBLEM SETUP.

We denote regular font letters as scalars, bold lower case letters as vectors and bold upper case letters as matrices. $a \lesssim b$ means $a \leq Cb$ for some constant $C$ that does not depend on $a$ or $b$, and $a \asymp b$ denotes $a \lesssim b$ and $b \lesssim a$. See Table 2 for the full list of symbols used.

Let $f_0$ be the target function to be estimated. The training dataset is $\mathcal{D}_n := \{(\boldsymbol{x}_i, y_i), y_i = f_0(\boldsymbol{x}_i) + \epsilon_i, i \in [n]\}$, where $x_i$ are fixed and $\epsilon_i$ are zero-mean, independent Gaussian noises with variance $\sigma^2$. In the following discussion, we assume $\boldsymbol{x}_i \in [0, 1]^d$, $f_0(x_i) \in [-1, 1], \forall i$.

We will be comparing estimators under the mean square error (MSE), defined as $\mathrm{MSE}(\hat{f}) := \mathbb{E}_{\mathcal{D}_n} \frac{1}{n} \sum_{i=1}^n (\hat{f}(\boldsymbol{x}_i) - f_0(\boldsymbol{x}_i))^2$. The optimal worst-case MSE is described by $R(\mathcal{F}) := \min_{\hat{f}} \max_{f_0 \in \mathcal{F}} \mathrm{MSE}(\hat{f})$. We say that $\hat{f}$ is optimal if $\mathrm{MSE}(\hat{f}) \lesssim R(\mathcal{F})$. The empirical (square error) loss is defined as $\hat{L}(\hat{f}) := \frac{1}{n} \sum_{i=1}^n (\hat{f}(\boldsymbol{x}_i) - y_i)^2$. The corresponding population loss is $L(\hat{f}) := \mathbb{E}[\frac{1}{n} \sum_{i=1}^n (\hat{f}(\boldsymbol{x}_i) - y_i')^2 | \hat{f}]$ where $y_i'$ are new data points. It is clear that $\mathbb{E}[L(\hat{f})] = \mathrm{MSE}[\hat{f}] + \sigma^2$.

### 2.2 BESOV SPACES AND BOUND VARIATION SPACE

**Besov space**, denoted as $B_{p,q}^\alpha$, is a flexible function class parameterized by $\alpha, p, q$ whose definition is deferred to Section C.1. Here $\alpha \geq 0$ determines the smoothness of functions, $1 \leq p \leq \infty$ determines the averaging (quasi-)norm over locations, $1 \leq q \leq \infty$ determines the averaging (quasi-)norm over scale which plays a relatively minor role. Smaller $p$ is more forgiving to inhomogeneity and loosely speaking, when the function domain is bounded, smaller $p$ induces a larger function space. On the other hand, it is easy to see from definition that $B_{p,q}^\alpha \subset B_{p,q'}^\alpha$, if $q < q'$. Without loss of generalizability, in the following discussion we will only focus on $B_{p,\infty}^\alpha$. When $p = 1$, the Besov space allows higher inhomogeneity, and it is more general than the Sobolev or Hölder space.

**Bounded variation (BV) space** is a more interpretable class of functions with spatially heterogeneous smoothness (Donoho et al., 1998). It is defined through the total variation (TV) of a function.

Table 2: Symbols used in this paper

| symbol | Meaning | | |
|---|---|---|---|
| $a/\boldsymbol{a}/\mathbf{A}$ | scalars / vectors / matrices. | $[a, b]$ | $\{x \in \mathbb{R} : a \leq x \leq b\}$ |
| $B_{p,q}^\alpha$ | Besov space. | $[n]$ | $\{x \in \mathbb{N} : 1 \leq x \leq n\}$. |
| $\|\cdot\|_{B_{p,q}^\alpha}$ | Besov quasi-norm . | $\|\cdot\|_F$ | Frobenius norm. |
| $\|\cdot\|_{B_{p,q}^\alpha}$ | Besov norm. | $\|\cdot\|_p$ | $\ell_p$-norm. |
| $M_m(\cdot)$ | $m^{th}$ order Cardinal B-spline bases. | $d$ | Dimension of input. |
| $M_{m,k,\boldsymbol{s}}(\cdot)$ | $m^{th}$ order Cardinal B-spline basis function of resolution $k$ at position $\boldsymbol{s}$. | $M$ | # subnetworks in a parallel NN. |
| | | $L$ | # layers in a (parallel) NN. |
| | | $w$ | Width of a subnetwork. |
| $\sigma(\cdot)$ | ReLU activation function. | $n$ | # samples. |
| $\mathbf{W}_j^{(\ell)}, \boldsymbol{b}_j^{(\ell)}$ | Weight and bias in the $\ell$-th layer in the $j$-th subnetwork. | $\mathbb{R}, \mathbb{Z}, \mathbb{N}$ | Set of real numbers, integers, and nonnegative integers. |

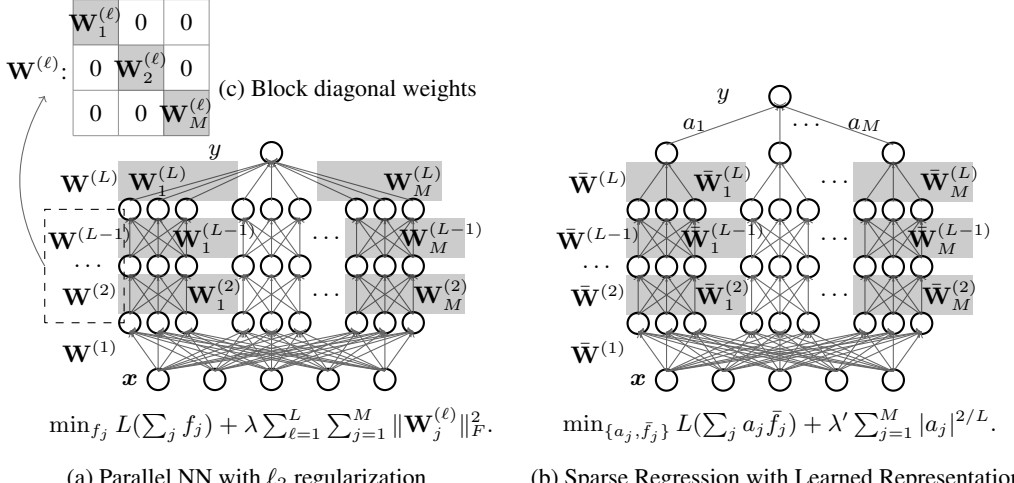

$$\min_{f_j} L(\sum_j f_j) + \lambda \sum_{\ell=1}^L \sum_{j=1}^M \|\mathbf{W}_j^{(\ell)}\|_F^2.$$

(a) Parallel NN with $\ell_2$ regularization

$$\min_{\{a_j, \bar{f}_j\}} L(\sum_j a_j \bar{f}_j) + \lambda' \sum_{j=1}^M |a_j|^{2/L}.$$

(b) Sparse Regression with Learned Representation

Figure 2: Parallel neural network and the equivalent sparse regression model we discovered.

For $(m+1)$th differentiable function $f : [0,1] \to \mathbb{R}$, the $m$th order total variation is defined as $TV^{(m)}(f) := TV(f^{(m+1)}) = \int_{[0,1]} |f^{(m+1)}(x)|dx$, and the corresponding $m$th order Bounded Variation class $BV(m) := \{f : TV(f^{(m)}) < \infty\}$. The more general definition is given in Section C.2. Bounded variation class is tightly connected to Besov classes. Specifically (DeVore & Lorentz, 1993):

$$B_{1,1}^{m+1} \subset BV(m) \subset B_{1,\infty}^{m+1} \tag{1}$$

This allows the results derived for the Besov space to be easily applied to BV space.

**Minimax MSE** It is well known that minimax rate for Besov and 1D BV classes are $O(n^{-\frac{2\alpha}{2\alpha+d}})$ and $O(n^{-(2m+2)/(2m+3)})$ respectively . The minimax rate for *linear estimators* in 1D BV classes is known to be $O(n^{-(2m+1)/(2m+2)})$ (Mammen & van de Geer, 1997; Donoho et al., 1998).

## 3 MAIN RESULTS: PARALLEL ReLU DNNs

Consider a parallel neural network containing $M$ multi layer perceptrons (MLP) with ReLU activation functions called *subnetworks*. Each subnetwork has width $w$ and depth $L$. The input is fed to all the subnetworks, and the output of the parallel NN is the summation of the output of each subnetwork. The architecture of a parallel neural network is shown in Figure 2a. This parallel neural network is equivalent to a vanilla neural network with block diagonal weights in all but the first and the last layers (Figure 2(c)). Let $\mathbf{W}_j^{(\ell)}$ and $\boldsymbol{b}_j^{(\ell)}$ denote the weight and bias in the $\ell$-th layer in the $j$-th subnetwork respectively. Training this model with $\ell_2$ regularization returns:

$$\underset{\{\mathbf{W}_j^{(\ell)}, \boldsymbol{b}_j^{(\ell)}\}}{\arg\min} \ \hat{L}(f) + \lambda \sum_{j=1}^M \sum_{\ell=1}^L \|\mathbf{W}_j^{(\ell)}\|_F^2, \tag{2}$$

where $f(x) = \sum_{j=1}^M f_k(x)$ denotes the parallel neural network, $f_j(\cdot)$ denotes the $j$-th subnetwork, and $\lambda > 0$ is a fixed scaling factor. We choose not to regularize the bias terms $\boldsymbol{b}_j^{(\ell)}$ to provide a cleaner equivalent model (Proposition 4). If the bias terms are regularized, the result will be similar. Besides, we ignore the computation issue and focus on the global optimal solution to this problem. In practice, in deep neural network, the solution obtained using gradient descent-style methods are often close to the global optimal solution (Choromanska et al., 2015).

**Theorem 1.** *For any fixed $\alpha - d/p > 1, q \geq 1, L \geq 3$, define $m = \lceil \alpha - 1 \rceil$. For any $f_0 \in B_{p,q}^\alpha$, given an $L$-layer parallel neural network satisfying*

- *The width of each subnetwork is **fixed** satisfying $w \geq O(md)$. See Theorem 8 for the detail.*

- *The number of subnetworks is **large enough**: $M \gtrsim n^{\frac{1-2/L}{2\alpha/d+1-2/(pL)}}$.*

*Under the assumption as in Lemma 17, with proper choice of the parameter of regularizaton $\lambda$ that depends on $\mathcal{D}, \alpha, d, L$, the solution $\hat{f}$ parameterized by (2) satisfies*

$$\text{MSE}(\hat{f}) = C(w, L)\tilde{O}\left(n^{-\frac{2\alpha/d(1-2/L)}{2\alpha/d+1-2/(pL)}}\right) + e^{-c_6 L}. \tag{3}$$

*where $\tilde{O}$ shows the scale up to a logarithmic factor, $c_6 > 0$ is a numerical constant from Theorem 8, $C(w, L) \asymp (w^{4-4/L}L^{2-4/L})^{\frac{2\alpha/d}{2\alpha/d+1-2/(pL)}}$ depends polynomially on $L$.*

We explain the proof idea in the next section, but defer the extended form of the theorem and the full proof to Section F. Before that, we comment on a few interesting aspects of the result.

**Near optimal rates and the effect of depth.** The first term in the MSE bound is the estimation error and the second term is (part of) the approximation error of this NN. Recall that the minimax rate of a Besov class is $O(n^{-\frac{2\alpha}{2\alpha+d}})$. The gap between the estimation error and the minimax rate is because the minimax rate can be achieved by an $\ell_0$ sparse model, while the parallel NN is equivalent to an $\ell_p$ sparse model (will be shown in Proposition 4), which is an approximation to $\ell_0$. As the depth parameter $L$ increases, $p = 2/L$ gets closer to 0, the MSE can get arbitrarily close to the minimax rate and the trailing constant term in (3) can be arbitrarily small. Close to the optimal rate can be achieved if we choose $L \gtrsim \log n$:

**Corollary 2.** *Under the conditions of Theorem 1, for any $f_0 \in B^\alpha_{p,q}$, there is a numerical constant $C$ such that when we choose $C \log n \le L \le 100 C \log n$,*

$$\text{MSE}(\hat{f}) = \tilde{O}(n^{-\frac{2\alpha}{2\alpha+d}(1-o(1))}),$$

*where $\tilde{O}$ hides only logarithmic factors and the $o(1)$ factor in the exponent is $O(1/\log(n))$.*

**Sparsity and comparison with standard NN.** We also note that the result does not depend on $M$ as long as $M$ is large enough. This means that the neural network can be arbitrarily overparameterized while not overfitting. The underlying reason is *sparsity*. As it will become clearer in Section 4.1, $\ell_2$ regularized training of a parallel $L$-layer ReLU NNs is equivalent to a sparse regression problem with an $\ell_p$ penalty assigned to the coefficient vector of a learned dictionary. Here $p = 2/L$ which promotes even sparser solutions than an $\ell_1$ penalty. Such $\ell_p$ sparsity does not exist in standard deep neural networks to the best of our knowledge, which indicates that parallel neural networks may be superior over standard neural networks in local adaptivity.

**Adaptivity to function spaces.** For any fixed $L, \tilde{m}$, our result shows the parallel neural network with width $w = O(\tilde{m}d)$ can achieve close to the minimax rate for any Besov class as long as $\alpha \le \tilde{m}$. In other words, neural networks can adapt to smoothness parameter by tuning only the regularizaton parameter. As will be shown in Theorem 5, overestimating $\alpha$ with $\tilde{m}$ only changes the logarithmic terms in the MSE bound — a mild price to pay for a more adaptive method.

**Hyperparameter tuning.** We provide an explicit choice of $\lambda$ in Lemma 17 underlying our theoretical result. Empirically, it can be determined empirically, e.g. using cross validation.

**Fixed design v.s. random design.** We mainly focus on bounding the error at sample covariates (the *fixed design* problem) to be comparable to classical nonparameteric regression results. For completeness, we also state results for the *random design* version of the problem (bounding $\mathbb{E}_\mathcal{D}\mathbb{E}_f\text{MSE}(\hat{f})$) in Theorem 19, to be compatible with the standard statistical learning setting (e.g., Suzuki, 2018).

More discussion about this result can be found in Section G.

**Bounded variation classes.** Thanks to the Besov space embedding of the BV class (1), our theorem also implies the result for the BV class in $1D$.

**Corollary 3.** *If the target function is in bounded variation class $f_0 \in BV(m)$, For any fixed $L \ge 3$, for a neural network satisfying the requirements in Theorem 1 with $d = 1$ and with proper choice of the regularization factor $\lambda$, the NN $\hat{f}$ parameterized by (6) satisfies*

$$\text{MSE}(\hat{f}) = C(w, L)\tilde{O}(n^{-\frac{(2m+2)(1-2/L)}{2m+3-2/L}}) + O(e^{-c_6 L}),$$

*where $C(w, L)$ is the same as in (3) except replacing $\alpha$ with $m$.*

It is known that any linear estimators such as kernel smoothing and smoothing splines cannot have an error lower than $O(n^{-(2m+1)/(2m+2)})$ for $BV(m)$ (Donoho et al., 1998). When $L > O(m^2)$, the first term in the MSE of NN decreases with $n$ faster than that of the linear methods. When $n$ is large enough, there exists $L$ such that the MSE of NN is strictly smaller than that of any linear method. This partly explains the advantage of DNNs over kernels.

## 4 PROOF OVERVIEW

We start by first proving that a parallel neural network trained with $\ell_2$ regularization is equivalent to an $\ell_p$-sparse regression problem with representation learning (Section 4.1); which helps decompose its MSE into an estimation error and approxmation error. Then we bound the two terms under an $\ell_p$-sparse constrained problem setting in Section 4.2 and Section 4.3 respectively.

Notably, we adapted the generic statistical learning machinery (a self-bounding argument) for studying this constrained ERM problem (Suzuki, 2018, Proposition 4) to bound the estimation error. This adaption is non-trival because there is an *unconstrained* subspace with no bounded metric entropy. Specifically, Proposition 15 shows that the MSE of the regression problem can be bounded by

$$\text{MSE}(\hat{f}) = O\bigg( \underbrace{\inf_{f \in \mathcal{F}} \text{MSE}(f)}_{\text{approximation error}} + \underbrace{\frac{\log \mathcal{N}(\mathcal{F}_\parallel, \delta, \|\cdot\|_\infty) + d(\mathcal{F}_\perp)}{n} + \delta}_{\text{estimation error}} \bigg) \tag{4}$$

in which $\mathcal{F}$ decomposes into $\mathcal{F}_\parallel \times \mathcal{F}_\perp$, where $\mathcal{F}_\perp$ is an unconstrained subspace with finite dimension, and $\mathcal{F}_\parallel$ is a compact set in the orthogonal complement with a $\delta$-covering number of $\mathcal{N}(\mathcal{F}_\parallel, \delta, \|\cdot\|_\infty)$ in $\|\cdot\|_\infty$-norm. This decomposes MSE into an approximation error and an estimation error. The novel analysis of these two represents the major technical contribution of this paper.

### 4.1 EQUIVALENCE TO $\ell_p$ SPARSE REGRESSION

It is widely known that ReLU function is 1-homogeneous: $\sigma(ax) = a\sigma(x), \forall a \geq 0, x \in \mathbb{R}$. In any consecutive two layers in a neural network (or a subnetwork), one can multiply the weight and bias in one layer with a positive constant, and divide the weight in another layer with the same constant. The neural network after such transformation is equivalent to the original one:

$$\mathbf{W}^{(2)}\sigma(\mathbf{W}^{(1)}\boldsymbol{x} + \boldsymbol{b}^{(1)}) = \frac{1}{c}\mathbf{W}^{(2)}\sigma(c\mathbf{W}^{(1)}\boldsymbol{x} + c\boldsymbol{b}^{(1)}), \quad \forall c > 0, \boldsymbol{x}. \tag{5}$$

This property can be applied to *each subnetwork* (instead of the entire model in a standard NN), and we can reformulate (2) to an $\ell_p$ sparsity-regularized problem:

**Proposition 4.** *There exists an one-to-one mapping between $\lambda > 0$ and $\lambda' > 0$ such that (2) is equivalent to the following problem:*

$$\underset{\{\bar{\mathbf{W}}_j^{(\ell)}, \bar{\boldsymbol{b}}_j^{(\ell)}, a_j\}}{\arg\min} \hat{L}\Big(\sum_{j=1}^{M} a_j \bar{f}_j\Big) + \lambda'\|\{a_j\}\|_{2/L}^{2/L} \tag{6}$$

$$s.t. \ \|\bar{\mathbf{W}}_j^{(1)}\|_F \leq c_1\sqrt{d}, \forall j \in [M]; \ \|\bar{\mathbf{W}}_j^{(\ell)}\|_F \leq c_1\sqrt{w}, \forall j \in [M], 2 \leq \ell \leq L,$$

*where $\bar{f}_j(\cdot)$ is a subnetwork with parameters $\bar{\mathbf{W}}_j^{(\ell)}, \bar{\boldsymbol{b}}_j^{(\ell)}$.*

This equivalent model is demonstrated in Figure 2b. The proof, which we defer to Section D.1, uses AM-GM inequality and the observation that the optimal solution will have norm-equalized weights per layer. The constraint $\|\bar{\mathbf{W}}_j^{(1)}\|_F \lesssim \sqrt{d}, \|\bar{\mathbf{W}}_j^{(\ell)}\|_F \lesssim \sqrt{w}, \forall \ell > 1$ is typical in deep learning for better numerical stability. The equivalent model in Proposition 4 is also a parallel neural network, but it appends one layer with parameters $\{a_k\}$ at the end of the neural network, and the constraint on the Frobenius norm is converted to the $2/L$ norm on the factors $\{a_k\}$. Since $L \gg 2$ in a typical application, $2/L \ll 1$ and this regularizer can enforce a sparser model than that in Section B. The same technique can also be used to prove that an $\ell_2$ constrained neural network is equivalent to the $\ell_{2/L}$ constrained model as in (7).

There are two useful implications of Proposition 4. First, it gives an intuitive explanation on how a regularized Parallel NN works. Specifically, it can be viewed as a sparse linear regression with representation learning. Secondly, the conversion into the constrained form allows us to decompose the MSE into two terms as in (4) and bound them separately.

We emphasize that Proposition 4 by itself is not new. The same result was previously obtained by Savarese et al. (2019, Appendix C) (see Section A for more details) and the key proof techniques date back to at least Burer & Monteiro (2003). Our novel contribution is to leverage this folklore equivalence for proving new learning bounds.

## 4.2 ESTIMATION ERROR ANALYSIS

Previous results that bound the covering number of neural networks (Yarotsky, 2017; Suzuki, 2018) depends on the width of the neural networks explicitly, which cannot be applied when analysing a potentially infinitely wide neural network. In this section, we leverage the $\ell_p$-norm bounded coefficients to avoid the dependence in $M$ in the covering number bound, and focus on a constrained optimization problem:

$$\underset{\{\bar{\mathbf{W}}_j^{(\ell)}, \bar{\boldsymbol{b}}_j^{(\ell)}, a_j\}}{\arg\min} \hat{L}\Big(\sum_{j=1}^{M} a_j \bar{f}_j\Big), \quad s.t. \|\{a_j\}\|_{2/L}^{2/L} \leq P', \tag{7}$$

and $\{\bar{\mathbf{W}}_j^{(\ell)}, \bar{\boldsymbol{b}}_j^{(\ell)}\}$ satisfy the same constraint as in (6). The connection between the regularized problem and the constrained problem is defered to Lemma 17.

**Theorem 5.** *The covering number of the model defined in (7) apart from the bias in the last layer satisfies*

$$\log \mathcal{N}(\mathcal{F}, \delta) \lesssim w^{2+2/(1-2/L)} L^2 \sqrt{d} P'^{\frac{1}{1-2/L}} \delta^{-\frac{2/L}{1-2/L}} \log(wP'/\delta). \tag{8}$$

This theorem provides a bound of estimation error for an arbitrarily wide parallel neural network as long as the total Frobenius norm is bounded. The proof can be found in Section D.2. It requires the following lemma, whose proof is deferred to Section D.3:

**Lemma 6.** *Let $\mathcal{G} \subseteq \{\mathbb{R}^d \to [-c_3, c_3]\}$ be a set with covering number satisfying $\log \mathcal{N}(\mathcal{G}, \delta) \lesssim k \log(1/\delta)$ for some finite $c_3$, and for any $g \in \mathcal{G}, |a| \leq 1$, we have $ag \in \mathcal{G}$. The covering number of $\mathcal{F} = \left\{\sum_{i=1}^{M} a_i g_i \Big| g_i \in \mathcal{G}, \|a\|_p^p \leq P, 0 < p < 1\right\}$ for any $P > 0$ satisfies*

$$\log \mathcal{N}(\mathcal{F}, \epsilon) \lesssim k P^{\frac{1}{1-p}} (\delta/c_3)^{-\frac{p}{1-p}} \log(c_3 P/\delta)$$

*up to a double logarithmic factor.*

## 4.3 APPROXIMATION ERROR ANALYSIS

The approximation error analysis involves two steps. We first analyse how a subnetwork can approximate a B-spline basis, which is defered to Section E.1. Then we show that a sparse linear combination of B-spline bases approximates Besov functions. Both add up to the total error in approximating Besov functions with a parallel neural network (Theorem 8).

**Proposition 7.** *Let $\alpha - d/p > 1, r > 0$. For any function in Besov space $f_0 \in B_{p,q}^\alpha$ and any positive integer $\bar{M}$, there is an $\bar{M}$-sparse approximation using B-spline basis of order $m$ satisfying $0 < \alpha < \min(m, m-1+1/p)$: $\check{f}_{\bar{M}} = \sum_{i=1}^{\bar{M}} a_{k_i, \boldsymbol{s}_i} M_{m, k_i, \boldsymbol{s}_i}$ for any positive integer $\bar{M}$ such that the approximation error is bounded as $\|\check{f}_{\bar{M}} - f_0\|_r \lesssim \bar{M}^{-\alpha/d} \|f_0\|_{B_{p,q}^\alpha}$, and the coefficients satisfy*

$$\|\{2^{k_i} a_{k_i, \boldsymbol{s}_i}\}_{k_i, \boldsymbol{s}_i}\|_p \lesssim \|f_0\|_{B_{p,q}^\alpha}.$$

The proof as well as the remark can be found in Section E.2.

**Theorem 8.** *Under the same condition as Proposition 7, for any positive integer $\bar{M}$, any function in Besov space $f_0 \in B_{p,q}^\alpha$ can be approximated by a parallel neural network with no more than $O(\bar{M})$ number of subnetworks satisfying:*

    *1. Each subnetwork has width $w = O(md)$ and depth $L$.*

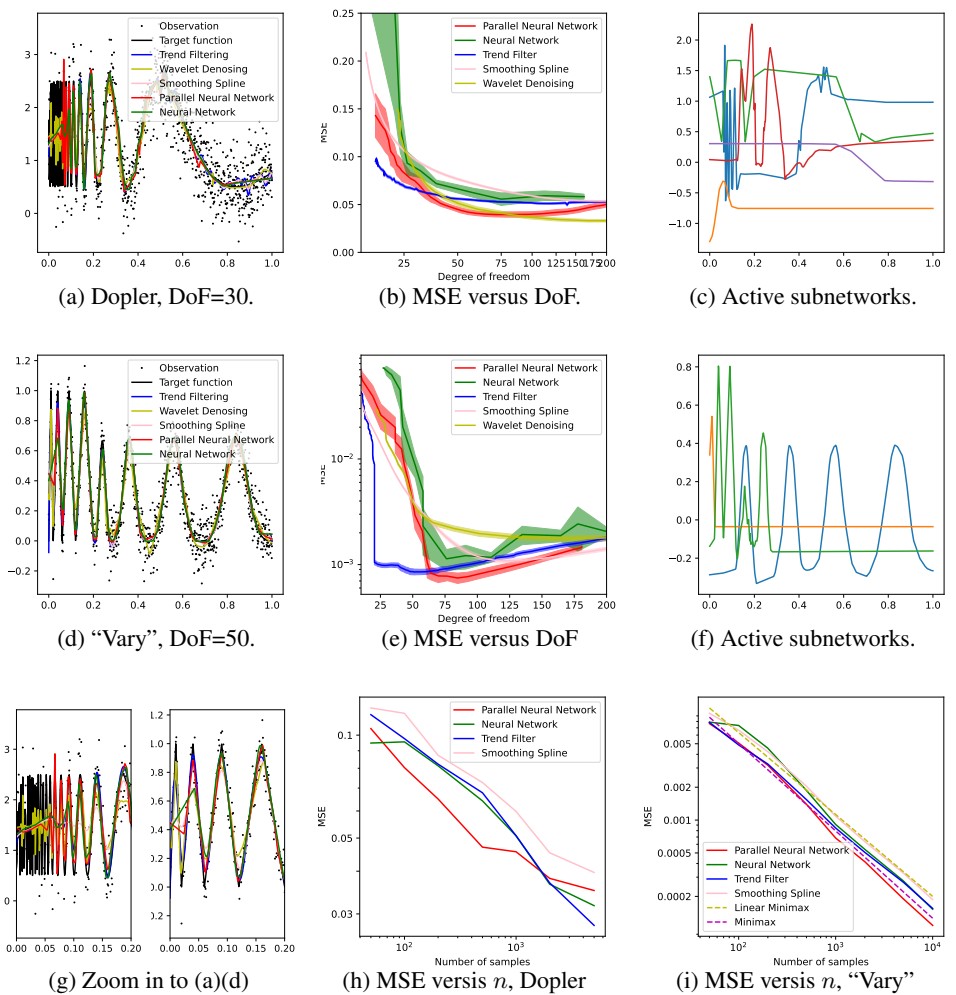

Figure 3: Numerical experiment results of the Doppler function (a-c,h), and "vary" function (d-f,g). All the "active" subnetworks are plotted in (c)(f). The horizontal axis in (b) is not linear.

2. *The weights in each layer satisfy $\|\bar{\mathbf{W}}_k^{(\ell)}\|_F \leq O(\sqrt{w})$ except the first layer $\|\bar{\mathbf{W}}_k^{(1)}\|_F \leq O(\sqrt{d})$,*

3. *The scaling factors have bounded $2/L$-norm: $P' := \|\{a_j\}\|_{2/L}^{2/L} \lesssim \bar{M}^{1-2/(pL)}$.*

4. *The approximation error is bounded by*

$$\|\tilde{f} - f_0\|_r \leq (c_4 \bar{M}^{-\alpha/d} + c_5 e^{-c_6 L})\|f\|_{B_{p,q}^\alpha}$$

*where $c_4, c_5, c_6$ are constants that depend only on $m, d$ and $p$.*

Here $\bar{M}$ is the number of "active" subnetworks, which is not to be confused with the number of subnetworks at initialization. The proof can be found in Section E.3.

Using the estimation error in Theorem 5 and approximation error in Theorem 8, by choosing $\bar{M}$ to minimax the total error, we can conclude the sample complexity of parallel neural networks using $\ell_2$ regularization, which is the main result (Theorem 1) of this paper. See Section F for the detail.

## 5 EXPERIMENT

We empirically compare a parallel neural network (PNN) and a vanilla ReLU neural network (NN) with smoothing spline, trend filtering (TF) (Tibshirani, 2014), and wavelet denoising. Trend filtering

can be viewed as a more efficient discrete spline version of locally adaptive regression spline and enjoys the same optimal rates for the BV classes. Wavelet denoising is also known to be minimax-optimal for the BV classes. The results are shown in Figure 3. We use two target functions: a Doppler function whose frequency is decreasing(Figure 3(a)-(c)(h)), and a combination of piecewise linear function and piecewise cubic function, or "vary" function (Figure 3(d)-(f)(i)). We repeat each experiment 10 times and take the average. The shallow area in Figure 3(b)(e) shows 95% confidence interval by inverting the Wald's test. The degree of freedom (DoF) is computed based on Tibshirani (2015).

As can be shown in the figure, both TF and wavelet denoising can adapt to the different levels of smoothness in the target function, while smoothing splines tend to be oversmoothed where the target function is less smooth (the left side in (a)(d), enlarged in (g)). The prediction of PNN is similar to TF and wavelet denoising and shows local adaptivity. Besides, the MSE of PNN almost follows the same trend as TF and wavelet denoising which is consistent with our theoretical understanding that the error rate of neural network is closer to locally adaptive methods. Notably PNN, TF and wavelet denoising achieve lower error at a much smaller degree-of-freedom than smoothing splines.

There are some mild drops in the best MSE one can achieve with Parallel NN vs TF in both examples. We are surprised that the drop is small because Parallel NN needs to learn the basis functions that TF essentially hard-coded. The additional price to pay for using a more adaptive and more flexible representation learning method seems not high at all.

In Figure 3(c)(f), we give the output *all* the "active" subnetwork, i.e. the subnetworks whose output is not a constant. Notice that the number of active subnetworks is much smaller than the initialization. This is because $\ell_2$ regularization in weights induces $\ell_p$ sparsity and the weight in most of the subnetworks reduces towards 0 after training. More details are shown in Section H.

In Figure 3(h)(i), we plot the MSE versus the number of training samples for "Doppler" and "Vary" respectively. It is clear that parallel NN works the best overall. In (i), we further compare the scaling of the MSE against the minimax rate ($n^{-4/5}$) and the minimax linear rate ($n^{-3/4}$), i.e., the best rate kernel methods could achieve. As is predicted by our theory, when $n$ is large, the MSE of parallel neural networks and trend filtering decreases at almost the same rate as the minimax rate, while smoothing splines, as expected, is converging at the (suboptimal) minimax linear rate. Interestingly, vanilla NN seems to converge at the optimal rate too on this example. It remains an open question whether vanila NN is merely "lucky" on this example, or it also achieves the minimax rate for all functions in BV(m).

## 6    Conclusion and Discussion

In this paper, we show that a deep parallel neural network can be locally adaptive with standard $\ell_2$ regularization. This confirms that neural networks can be nearly optimal in learning functions with heterogeneous smoothness which separates them from kernel methods.

Specifically, we prove that training an $L$ layer parallel neural network with standard $\ell_2$ regularization is equivalent to an $\ell_{2/L}$-penalized regression model with representation learning. Since in typical application $L \gg 2$, standard regularization promotes a sparse linear combination of the learned bases. Using this method, we proved that a parallel neural network can achieve close to the minimax rate in the Besov space and bounded variation (BV) space by tuning the regularization factor.

Our result reveals that one do not need to specify the smoothness parameter $\alpha$ (or $m$) when training a parallel neural network. With only an estimation of the upper bound of $\alpha$ (or $m$), parallel neural networks can adapt to different degree of smoothness, or choose different parameters for different regions of the domain of the target function. This property shows the strong adaptivity of deep neural networks.

On the other hand, as the depth of neural network $L$ increases, $2/L$ tends to 0 and the error rate moves closer to the minimax rate of Besov and BV space. This indicates that when the sample size is large enough, deeper models have smaller error than shallower models, and helps explain why empirically deep neural networks has better performance than shallow neural networks.

## ACKNOWLEDGMENTS

The work is partially supported by NSF Award #2134214. The authors thank Alden Green for references on Besov-space embedding of BV classes, Dheeraj Baby for helpful discussion on B-splines as well as Ryan Tibshirani on connections to (Tibshirani, 2021) and for sharing with us a simpler proof of the Theorem 1 of (Parhi & Nowak, 2021a) based on Caratheorodory's Theorem (used in the proof of Theorem 9 on two-layer NNs).

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

## A  OTHER RELATED WORKS

**NN and kernel methods.**  Jacot et al. (2018) draws the connection between neural networks and kernel methods. However, it has been found that neural networks often outperform any kernel method, especially when the learning rate is relatively large (Lewkowycz et al., 2020). A series of work tried to distinguish NN from kernel methods by providing examples of function spaces that NN provably outperform kernel methods (Allen-Zhu & Li, 2019; Ghorbani et al., 2020). However, these papers did not consider the local adaptivity of nerual networks, which provides a more systematic explanation.

**NN and splines.**  Besides Parhi & Nowak (2021a) which we discussed earlier, Parhi & Nowak (2021b;c) also leveraged the connections between NNs and splines. Parhi & Nowak (2021b) focused on characterizing the variational form of multi-layer NN. Parhi & Nowak (2021c) showed that two-layer ReLU activated NN achieves minimax rate for a BV class of order $1$ but did not cover multilayer NNs nor BV class with order $> 1$, which is our focus.

**Weight-decay regularization with sparsity-inducing penalties.**  The connection between weight-decay regularization with sparsity-inducing penalties in two-layer NNs is folklore and used by Neyshabur et al. (2014); Savarese et al. (2019); Ongie et al. (2019); Ergen & Pilanci (2021a;d); Parhi & Nowak (2021a;c); Pilanci & Ergen (2020). The key underlying technique — an application of the AM-GM inequality (which we used in this paper as well) — can be traced back to Srebro et al. (2004) (see a recent exposition by Tibshirani (2021)). Tibshirani (2021) also generalized the result to multi-layered NNs, but with a simple (element-wise) connections. Besides, Ergen & Pilanci (2020) proved that training a two-layer convolution neural network (CNN) with weight decay induces sparsity, and points to a potential extension to these works including our work.

Finally, it was brought to our attention that while Savarese et al. (2019) mainly consider two-layer NNs, a set of results about $L$-layer parallel NNs was presented in Appendix C of their paper, which essentially contains same arguments we used for proving the equivalence to an $\ell_{2/L}$ regularized optimization problem in Proposition 4. The difference is they applied the insight to understand the interpolation regime while we focused on analyzing MSE in the noisy case.

Proposition 4, is the Savarese et al. (2019) showed that a parallel networks of depth $L$ have an inductive bias for the $L_{2/L}$ sparse model, and explicit weight decay causes the solutions of these networks to have a sparse last layer with at most $n$ nonzero weights.

**Resnet-type convolution neural networks.**  A recent series of work (Oono & Suzuki, 2019; Liu et al., 2021) proves that an arbitrary parallel neural network can be approximated by a resnet-type convolution neural networks. These works do not require the model to be sparse, thus are easier to train, yet they still require the architecture (the width and depth of each residual block, the number of residual blocks) to be tuned based on the dataset, and the estimation error analysis is based on the number of parameters. Besides, the number of residual block need to increase with $n$, making the entire too deep to train in practice.

**Approximation and estimation.**  The approximation-theoretic and estimation-theoretic research for neural network has a long history too (Cybenko, 1989; Barron, 1994; Yarotsky, 2017; Schmidt-Hieber, 2020; Suzuki, 2018). Most existing work considered the Holder, Sobolev spaces and their extensions, which contain only homogeneously smooth functions and cannot demonstrate the advantage of NNs over kernels.  The exceptions including Suzuki (2018); Oono & Suzuki (2019); Liu et al. (2021) which, as we discussed earlier, requires modifications to NN architecture for each class. In contrast, we require tuning only the standard weight decay parameter. Most importantly, in all previously works, the estimation error of the model (eg. the covering number) depends on the number of nonzero parameters in the model, while our work provides a bound that depends on the norm of the weights instead of the number of subnetworks.

## B  TWO-LAYER NEURAL NETWORK WITH TRUNCATED POWER ACTIVATION FUNCTIONS

We start by recapping the result of Parhi & Nowak (2021a) and formalizing its implication in estimating BV functions. Parhi & Nowak (2021a) considered a two layer neural network with truncated

power activation function. Let the neural network be

$$f(x) = \sum_{j=1}^{M} v_j \sigma^m(w_j x + b_j) + c(x), \tag{9}$$

where $w_j, v_j$ denote the weight in the first and second layer respectively, $b_j$ denote the bias in the first layer, $c(x)$ is a polynomial of order up to $m$, $\sigma^m(x) := \max(x, 0)^m$. Parhi & Nowak (2021a, Theorem 8) showed that when $M$ is large enough, The optimization problem

$$\min_{\boldsymbol{w}, \boldsymbol{v}} \hat{L}(f) + \frac{\lambda}{2} \sum_{j=1}^{M} (|v_j|^2 + |w_j|^{2m}) \tag{10}$$

is equivalent to the locally adaptive regression spline:

$$\min_{f} \hat{L}(f) + \lambda TV(f^{(m)}(x)), \tag{11}$$

which optimizes over arbitrary functions that is $m$-times weakly differentiable. The latter was studied in Mammen & van de Geer (1997), which leads to the following MSE:

**Theorem 9.** *Let $M \geq n - m$, and $\hat{f}$ be the function (9) parameterized by the minimizer of (10), then*

$$\text{MSE}(\hat{f}) = O(n^{-(2m+2)(2m+3)}).$$

We show a simpler proof in the univariate case due to Tibshirani (2022):

*Proof.* As is shown in Parhi & Nowak (2021a, Theorem 8), the minimizer of (10) satisfy

$$|v_j| = |w_j|^m, \forall k$$

so the TV of the neural network $f_{NN}$ is

$$TV^{(m)}(f_{NN}) = TV^{(m)} c(x) + \sum_{j=1}^{M} |v_j| |w_j|^m TV^{(m)}(\sigma^{(m)}(x))$$

$$= \sum_{j=1}^{M} |v_j| |w_j|^m$$

$$= \frac{1}{2} \sum_{j=1}^{M} (|v_j|^2 + |w_j|^{2m})$$

which shown that (10) is equivalent to the locally adaptive regression spline (11) as long as the number of knots in (11) is no more than $M$. Furthermore, it is easy to check that any spline with knots no more than $M$ can be expressed as a two layer neural network (10). It suffices to prove that the solution in (11) has no more than $n - m$ number of knots.

Mammen & van de Geer (1997, Proposition 1) showed that there is a solution to (11) $\hat{f}(x)$ such that $\hat{f}(x)$ is a $m$th order spline with a finite number of knots but did not give a bound. Let the number of knots be $M$, we can represent $\hat{f}$ using the truncated power basis

$$\hat{f}(x) = \sum_{j=1}^{M} a_j (x - t_j)_+^m + c(x) := \sum_{j=1}^{M} a_j \sigma_j^{(m)}(x) + c(x)$$

where $t_j$ are the knots, $c(x)$ is a polynomial of order up to $m$, and define $\sigma_j^{(m)}(x) = (x - t_j)_+^m$.

Mammen & van de Geer (1997) however did not give a bound on $M$. Parhi & Nowak (2021a)'s Theorem 1 implies that $M \leq n - m$. Its proof is quite technical and applies more generally to a higher dimensional generalization of the BV class.

Tibshirani (2022) communicated to us the following elegant argument to prove the same using elementary convex analysis and linear algebra, which we present below.

Define $\Pi_m(f)$ as the $L^2(P_n)$ projection of $f$ onto polynomials of degree up to $m$, $\Pi_m^\perp(f) := f - \Pi_m(f)$. It is easy to see that

$$\Pi_m^\perp f(x) = \sum_{j=1}^M a_j \Pi_m^\perp \sigma_j^{(m)}(x)$$

Denote $f(x_{1:n}) := \{f(x_1), \ldots, f(x_n)\} \in \mathbb{R}^n$ as a vector of all the predictions at the sample points.

$$\Pi_m^\perp \hat{f}(x_{1:n}) = \sum_{j=1}^M a_j \Pi_m^\perp \sigma_j^{(m)}(x_{1:n}) \in \Pi_m^\perp \text{conv}\{\pm \sigma_j^{(m)}(x_{1:n})\} \cdot \sum_{j=1}^M |a_j|$$

$$\in \text{conv}\{\pm \Pi_m^\perp \sigma_j^{(m)}(x_{1:n})\} \cdot \sum_{j=1}^M |a_j|$$

where conv denotes the convex hull of a set. The convex hull $\text{conv}\{\pm \sigma_j^{(m)}(x_{1:n})\} \cdot \sum_{j=1}^M |a_j|$ is an $n$-dimensional space, and polynomials of order up to $m$ is an $m+1$ dimensional space, so the set defined above has dimension $n - m - 1$. By Carathéodory's theorem, there is a subset of points in this space

$$\{\Pi_m^\perp \sigma_{j_k}^{(m)}(x_{1:n})\} \subseteq \{\Pi_m^\perp \sigma_j^{(m)}(x_{1:n})\}, 1 \leq k \leq n - m$$

such that

$$\Pi_m^\perp f(x) = \sum_{k=1}^{n-m} \tilde{a}_k \Pi_m^\perp \sigma_{j_k}^{(m)}(x), \sum_{k=1}^{n-m} |a_k| \leq 1$$

In other word, there exist a subset of knots $\{\tilde{t}_j, j \in [n-m]\}$ that perfectly recovers $\Pi_m^\perp \hat{f}(x)$ at all the sample points, and the TV of this function is no larger than $\hat{f}$.

This shows that

$$\tilde{f}(x) = \sum_{j=1}^{n-m} \tilde{a}_j (x - t_j)_+^m, s.t. \tilde{f}(x_i) = f(x_i)$$

for all $x_i$ in $n$ onbservation points.

The MSE of locally adaptivity regressive spline (11) was studied in Mammen & van de Geer (1997, Section 3), which equals the error rate given in Theorem 9. $\qquad \square$

This indicates that the neural network (9) is minimax optimal for $BV(m)$.

Let us explain a few the key observations behind this equivalence. (a) The truncated power functions (together with an $m$th order polynomial) spans the space of an $m$th order spline. (b) The neural network in (9) is equivalent to a free-knot spline with $M$ knots (up to reparameterization). (c) A solution to (11) is a spline with at most $n - m$ knots (Parhi & Nowak, 2021a, Theorem 8). (d) Finally, by the AM-GM inequality

$$|v_j|^2 + |w_j|^{2m} \geq 2|v_j||w_j|^m = 2|c_j|$$

where $c_j = v_j|w_j|^m$ is the coefficient of the corresponding $j$th truncated power basis. The $m$th order total variation of a spline is equal to $\sum_j |c_j|$. It is not hard to check that the loss function depends only on $c_j$, thus the optimal solution will always take "=" in the AM-GM inequality.

## C    INTRODUCTION TO COMMON FUNCTION CLASSES

In the following definition define $\Omega$ be the domain of the function classes, which will be omitted in the definition.

### C.1 BESOV CLASS

**Definition 1.** *Modulus of smoothness: For a function $f \in L^p(\Omega)$ for some $1 \leq p \leq \infty$, the r-th modulus of smoothness is defined by*

$$w_{r,p}(f,t) = \sup_{h \in \mathbb{R}^d : \|h\|_2 \leq t} \|\Delta_h^r(f)\|_p,$$

$$\Delta_h^r(f) := \begin{cases} \sum_{j=0}^r \binom{r}{j}(-1)^{r-j} f(x+jh), & \text{if } x \in \Omega, x + rh \in \Omega, \\ 0, & \text{otherwise.} \end{cases}$$

**Definition 2.** *Besov space: For $1 \leq p, q \leq \infty, \alpha > 0, r := \lceil \alpha \rceil + 1$, define*

$$|f|_{B_{p,q}^\alpha} = \begin{cases} \left( \int_{t=0}^\infty (t^{-\alpha} w_{r,p}(f,t))^q \frac{dt}{t} \right)^{\frac{1}{q}}, & q < \infty \\ \sup_{t>0} t^{-\alpha} w_{r,p}(f,t), & q = \infty, \end{cases}$$

*and define the norm of Besov space as:*

$$\|f\|_{B_{p,q}^\alpha} = \|f\|_p + |f|_{B_{p,q}^\alpha}.$$

*A function $f$ is in the Besov space $B_{p,q}^\alpha$ if $\|f\|_{B_{p,q}^\alpha}$ is finite.*

Note that the Besov space for $0 < p, q < 1$ is also defined, but in this case it is a quasi-Banach space instead of a Banach space and will not be covered in this paper.

Functions in Besov space can be decomposed using B-spline basis functions. Any function $f$ in Besov space $B_{p,q}^\alpha, \alpha > d/p$ can be decomposed using B-spline of order $m, m > \alpha$: let $\boldsymbol{x} \in \mathbb{R}^d$,

$$f(\boldsymbol{x}) = \sum_{k=0}^\infty \sum_{\boldsymbol{s} \in J(k)} c_{k,\boldsymbol{s}}(f) M_{m,k,\boldsymbol{s}}(\boldsymbol{x}) \tag{12}$$

where $J(k) := \{2^{-k}\boldsymbol{s} : \boldsymbol{s} \in [-m, 2^k + m]^d \subset \mathbb{Z}^d\}$, $M_{m,k,\boldsymbol{s}}(\boldsymbol{x}) := M_m(2^k(\boldsymbol{x}-\boldsymbol{s}))$, and $M_k(\boldsymbol{x}) = \prod_{i=1}^d M_k(x_i)$ is the cardinal B-spline basis function which can be expressed as a polynomial:

$$
\begin{aligned}
M_m(x) &= \frac{1}{m!} \sum_{j=1}^{m+1} (-1)^j \binom{m+1}{j} (x-j)_+^m \\
&= ((m+1)/2)^m \frac{1}{m!} \sum_{j=1}^{m+1} (-1)^j \binom{m+1}{j} \left( \frac{x-j}{(m+1)/2} \right)_+^m,
\end{aligned}
\tag{13}
$$

Furthermore, the norm of Besov space is equivalent to the sequence norm:

$$\|\{c_{k,\boldsymbol{s}}\}\|_{b_{p,q}^\alpha} := \left( \sum_{k=0}^\infty (2^{(\alpha-d/p)k} \|\{c_{k,\boldsymbol{s}}(f)\}_{\boldsymbol{s}}\|_p)^q \right)^{1/q} \asymp \|f\|_{B_{p,q}^\alpha}.$$

See e.g. Dũng (2011, Theorem 2.2) for the proof.

The Besov space is closely connected to other function spaces including the Hölder space ($\mathcal{C}^\alpha$) and the Sobolev space ($W_p^\alpha$). Specifically, if the domain of the functions is $d$-dimensional (Suzuki, 2018; Sadhanala et al., 2021),

- $\forall \alpha \in \mathbb{N}, B_{p,1}^\alpha \subset W_p^\alpha \subset B_{p,\infty}^\alpha$, and $B_{2,2}^\alpha = W_2^\alpha$.
- For $0 < \alpha < \infty$ and $\alpha \in \mathcal{N}, \mathcal{C}^\alpha = B_{\infty,\infty}^\alpha$.
- If $\alpha > d/p, B_{p,q}^\alpha \subset \mathcal{C}^0$.

## C.2 Other Function Spaces

**Definition 3.** *Hölder space: let $m \in \mathbb{N}$, the $m$-th order Holder class is defined as*

$$\mathcal{C}^m = \left\{ f : \max_{|a|=k} \frac{|D^a f(x) - D^a f(z)|}{\|x - z\|_2} < \infty, \forall x, z \in \Omega \right\}$$

*where $D^a$ denotes the weak derivative.*

Note that fraction order of Hölder space can also be defined. For simplicity, we will not cover that case in this paper.

**Definition 4.** *Sobolev space: let $m \in \mathcal{N}, 1 \le p \le \infty$, the Sobolev norm is defined as*

$$\|f\|_{W_p^m} := \left( \sum_{|a| \le m} \|D^a f\|_p^p \right)^{1/p},$$

*the Sobolev space is the set of functions with finite Sobolev norm:*

$$W_p^m := \{ f : \|f\|_{W_p^m} < \infty \}.$$

**Definition 5.** *Total Variation (TV): The total variation (TV) of a function $f$ on an interval $[a, b]$ is defined as*

$$TV(f) = \sup_{\mathcal{P}} \sum_{i=1}^{n_{\mathcal{P}}-1} |f(x_{i+1}) - f(x_i)|$$

*where the $\mathcal{P}$ is taken among all the partitions of the interval $[a, b]$.*

In many applications, functions with stronger smoothness conditions are needed, which can be measured by high order total variation.

**Definition 6.** *High order total variation: the $m$-th order total variation is the total variation of the $(m-1)$-th order derivative*

$$TV^{(m)}(f) = TV(f^{(m-1)})$$

**Definition 7.** *Bounded variation (BV): The $m$-th order bounded variation class is the set of functions whose total variation (TV) is bounded.*

$$BV(m) := \{ f : TV(f^{(m)}) < \infty \}.$$

## D   Proof of Estimation Error

### D.1   Equivalence Between Parallel Neural Networks and $p$-norm Penalized Problems

**Proposition 4.** *There exists an one-to-one mapping between $\lambda > 0$ and $\lambda' > 0$ such that (2) is equivalent to the following problem:*

$$\underset{\{\bar{\mathbf{W}}_j^{(\ell)}, \bar{\boldsymbol{b}}_j^{(\ell)}, a_j\}}{\arg\min} \hat{L}\left( \sum_{j=1}^M a_j \bar{f}_j \right) + \lambda' \|\{a_j\}\|_{2/L}^{2/L}$$

$$s.t. \|\bar{\mathbf{W}}_j^{(1)}\|_F \le c_1 \sqrt{d}, \forall j \in [M]; \ \|\bar{\mathbf{W}}_j^{(\ell)}\|_F \le c_1 \sqrt{w}, \forall j \in [M], 2 \le \ell \le L,$$

*where $\bar{f}_j(\cdot)$ is a subnetwork with parameters $\bar{\mathbf{W}}_j^{(\ell)}, \bar{\boldsymbol{b}}_j^{(\ell)}$.*

*Proof.* We make use of the property from (5) to minimize the constraint term in (7) while keeping this neural network equivalent to the original one. Specifically, let $\mathbf{W}^{(1)}, \boldsymbol{b}^{(1)}, \dots \mathbf{W}^{(L)}, \boldsymbol{b}^{(L)}$ be the parameters of an $L$-layer neural network.

$$f(x) = \mathbf{W}^{(L)} \sigma(\mathbf{W}^{(L-1)} \sigma(\dots \sigma(\mathbf{W}^{(1)} x + \boldsymbol{b}^{(1)}) \dots) + \boldsymbol{b}^{(L-1)}) + \boldsymbol{b}^{(L)},$$

which is equivalent to

$$f(x) = \alpha_L \tilde{\mathbf{W}}^{(L)} \sigma(\alpha_{L-1} \tilde{\mathbf{W}}^{(L-1)} \sigma(\ldots \sigma(\alpha_1 \tilde{\mathbf{W}}^{(1)} x + \tilde{\boldsymbol{b}}^{(1)}) \ldots) + \tilde{\boldsymbol{b}}^{(L-1)}) + \tilde{\boldsymbol{b}}^{(L)},$$

as long as $\alpha_\ell > 0, \prod_{\ell=1}^L \alpha^L = \prod_{\ell=1}^L \|\mathbf{W}^{(\ell)}\|_F$, where $\tilde{\mathbf{W}}^{(\ell)} := \frac{\mathbf{W}^{(\ell)}}{\|\mathbf{W}^{(\ell)}\|_F}$. By the AM-GM inequality, the $\ell_2$ regularizer of the latter neural network is

$$\sum_{\ell=1}^L \|\alpha_\ell \tilde{\mathbf{W}}^{(\ell)}\|_F^2 = \sum_{\ell=1}^L \alpha_\ell^2 \geq L \left( \prod_{\ell=1}^L a_\ell \right)^{2/L} = L \left( \prod_{\ell=1}^L \|\mathbf{W}^{(\ell)}\|_F \right)^{2/L}$$

and equality is reached when $\alpha_1 = \alpha_2 = \cdots = \alpha_L$. In other word, in the problem (2), it suffices to consider the network that satisfies

$$\|\mathbf{W}_j^{(1)}\|_F = \|\mathbf{W}_j^{(2)}\|_F = \cdots = \|\mathbf{W}_j^{(L)}\|_F, \forall j \in [M], \ell \in [L]. \tag{14}$$

Using (5) again, one can find that the neural network is also equivalent to

$$f(x) = \sum_{j=1}^M a_j \bar{\mathbf{W}}^{(L)} \sigma(\bar{\mathbf{W}}_j^{(L-1)} \sigma(\ldots \sigma(\bar{\mathbf{W}}_j^{(1)} x + \bar{\boldsymbol{b}}_j^{(1)}) \ldots) + \bar{\boldsymbol{b}}_j^{(L-1)}) + \bar{\boldsymbol{b}}_j^{(L)},$$

where

$$\|\bar{\mathbf{W}}_j^{(\ell)}\|_F \leq \beta^{(\ell)}, a_j = \frac{\prod_{\ell=1}^L \|\mathbf{W}_j^{(\ell)}\|_F}{\prod_{\ell=1}^L \beta^{(\ell)}} = \frac{\|\mathbf{W}_j^{(1)}\|_F^L}{\prod_{\ell=1}^L \beta^{(\ell)}} = \frac{(\sum_{\ell=1}^L \|\mathbf{W}_j^{(\ell)}\|_F^2 / L)^{L/2}}{\prod_{\ell=1}^L \beta^{(\ell)}}, \tag{15}$$

where the last two equality comes from the assumption (14). Choosing $\beta^{(\ell)} = c_1 \sqrt{w}$ expect $\ell = 1$ where $\beta^{(1)} = c_1 \sqrt{d}$, and scaling $\bar{\boldsymbol{b}}^{(\ell)}$ accordingly and taking the regularizer in (2) into (15) finishes the proof. $\qquad \square$

## D.2 Covering Number of Parallel Neural Networks

**Theorem 5.** *The covering number of the model defined in (7) apart from the bias in the last layer satisfies*

$$\log \mathcal{N}(\mathcal{F}, \delta) \lesssim w^{2+2/(1-2/L)} L^2 \sqrt{d} P'^{\frac{1}{1-2/L}} \delta^{-\frac{2/L}{1-2/L}} \log(wP'/\delta).$$

The proof relies on the covering number of each subnetwork in a parallel neural network (Lemma 10), observing that $|f(x)| \leq 2^{L-1} w^{L-1} \sqrt{d}$ under the condition in Lemma 10, and then apply Lemma 6. We argue that our choice of condition on $\|\boldsymbol{b}^{(\ell)}\|_2$ in Lemma 10 is sufficient to analyzing the model apart from the bias in the last layer, because it guarantees that $\sqrt{w}\|\mathbf{W}^{(\ell)} \mathcal{A}_{\ell-1}(x)\|_2 \leq \|\boldsymbol{b}^{(\ell)}\|_2$. This leads to

$$\|\mathbf{W}^{(\ell)} \mathcal{A}_{\ell-1}(\boldsymbol{x})\|_\infty \leq \|\mathbf{W}^{(\ell)} \mathcal{A}_{\ell-1}(\boldsymbol{x})\|_2 \leq \sqrt{w}\|\boldsymbol{b}^{(\ell)}\|_2 \leq \|\boldsymbol{b}^{(\ell)}\|_\infty$$

If this condition is not met, $\mathbf{W}^{(\ell)} \mathcal{A}_{\ell-1}(\boldsymbol{x}) + b^{(\ell)}$ is either always positive or always negative for all feasible $\boldsymbol{x}$ along at least one dimension. If $(\mathbf{W}^{(\ell)} \mathcal{A}_{\ell-1}(\boldsymbol{x}) + b^{(\ell)})_i$ is always negative, one can replace $b^{(\ell)})_i$ with $-\max_{\boldsymbol{x}} \|\mathbf{W}^{(\ell)} \mathcal{A}_{\ell-1}(\boldsymbol{x})\|_\infty$ without changing the output of this model for any feasible $\boldsymbol{x}$. If $(\mathbf{W}^{(\ell)} \mathcal{A}_{\ell-1}(\boldsymbol{x}) + b^{(\ell)})_i$ is always positive, one can replace $b^{(\ell)})_i$ with $\max_{\boldsymbol{x}} \|\mathbf{W}^{(\ell)} \mathcal{A}_{\ell-1}(\boldsymbol{x})\|_\infty$, and adjust the bias in the next layer such that the output of this model is not changed for any feasible $\boldsymbol{x}$. In either cases, one can replace the bias $\boldsymbol{b}^{(\ell)}$ with another one with smaller norm while keeping the model equivalent except the bias in the last layer.

**Lemma 10.** *Let $\mathcal{F} \subseteq \{f : R^d \to \mathbb{R}\}$ denote the set of $L$-layer neural network (or a subnetwork in a parallel neural network) with width $w$ in each hidden layer. It has the form*

$$f(x) = \mathbf{W}^{(L)} \sigma(\mathbf{W}^{(L-1)} \sigma(\ldots \sigma(\mathbf{W}^{(1)} x + \boldsymbol{b}^{(1)}) \ldots) + \boldsymbol{b}^{(L-1)}) + \boldsymbol{b}^{(L)},$$
$$\mathbf{W}^{(1)} \in \mathbb{R}^{w \times d}, \|\mathbf{W}^{(1)}\|_F \leq \sqrt{d}, \boldsymbol{b}^{(1)} \in \mathbb{R}^w, \|\boldsymbol{b}^{(1)}\|_2 \leq \sqrt{dw},$$
$$\mathbf{W}^{(\ell)} \in \mathbb{R}^{w \times w} \|\mathbf{W}^{(\ell)}\|_F \leq \sqrt{w}, \boldsymbol{b}^{(\ell)} \in \mathbb{R}^w, \|\boldsymbol{b}^{(\ell)}\|_2 \leq 2^{\ell-1} w^{\ell-1} \sqrt{dw}, \quad \forall \ell = 2, \ldots L-1,$$
$$\mathbf{W}^{(L)} \in \mathbb{R}^{1 \times w}, \|\mathbf{W}^{(L)}\|_F \leq \sqrt{w}, b^{(L)} = 0 \tag{16}$$

*and $\sigma(\cdot)$ is the ReLU activation function, the input satisfy $\|x\|_2 \leq 1$, then the supremum norm $\delta$-covering number of $\mathcal{F}$ obeys*

$$\log \mathcal{N}(\mathcal{F}, \delta) \leq c_7 L w^2 \log(1/\delta) + c_8$$

*where $c_7$ is a constant depending only on $d$, and $c_8$ is a constant that depend on $d, w$ and $L$.*

*Proof.* First study two neural networks which differ by only one layer. Let $g_\ell, g'_\ell$ be two neural networks satisfying (16) with parameters $\mathbf{W}_1, \boldsymbol{b}_1, \ldots, \mathbf{W}_L, \boldsymbol{b}_L$ and $\mathbf{W}'_1, \boldsymbol{b}'_1, \ldots, \mathbf{W}'_L, \boldsymbol{b}'_L$ respectively. Furthermore, the parameters in these two models are the same except the $\ell$-th layer, which satisfy

$$\|\mathbf{W}_\ell - \mathbf{W}'_\ell\|_F \leq \epsilon, \|\boldsymbol{b}_\ell - \boldsymbol{b}'_\ell\|_2 \leq \tilde{\epsilon}.$$

Denote the model as

$$g_\ell(x) = \mathcal{B}_\ell(\mathbf{W}_\ell \mathcal{A}_\ell(\boldsymbol{x}) + \boldsymbol{b}_\ell), g'_\ell(x) = \mathcal{B}_\ell(\mathbf{W}'_\ell \mathcal{A}_\ell(\boldsymbol{x}) + \boldsymbol{b}'_\ell)$$

where $\mathcal{A}_\ell(\boldsymbol{x}) = \sigma(\mathbf{W}_{\ell-1}\sigma(\ldots \sigma(\mathbf{W}_1 x + \boldsymbol{b}_1)\ldots) + \boldsymbol{b}_{\ell-1})$ denotes the first $\ell-1$ layers in the neural network, and $\mathcal{A}_\ell(x) = \mathbf{W}_L \sigma(\ldots \sigma(\mathbf{W}_{\ell+1}\sigma(x) + \boldsymbol{b}_{\ell+1})\ldots) + \boldsymbol{b}_L$ denotes the last $L - \ell - 1$ layers, with definition $\mathcal{A}_1(\boldsymbol{x}) = \boldsymbol{x}, \mathcal{B}_L(\boldsymbol{x}) = \boldsymbol{x}$.

Now focus on bounding $\|\mathcal{A}(\boldsymbol{x})\|$. Let $\mathbf{W} \in \mathbb{R}^{m \times m'}, \|\mathbf{W}\|_F \leq \sqrt{m'}, \boldsymbol{x} \in \mathbb{R}^{m'}, \boldsymbol{b} \in \mathbb{R}^m, \|\boldsymbol{b}\|_2 \leq \sqrt{m}$

$$\begin{aligned}
\|\sigma(\mathbf{W}x + \boldsymbol{b})\|_2 &\leq \|\mathbf{W}\boldsymbol{x} + \boldsymbol{b}\|_2 \\
&\leq \|\mathbf{W}\|_2 \|\boldsymbol{x}\|_2 + \|\boldsymbol{b}\|_2 \\
&\leq \|\mathbf{W}\|_F \|\boldsymbol{x}\|_2 + \|\boldsymbol{b}\|_2 \\
&\leq \sqrt{m'}\|\boldsymbol{x}\|_2 + \sqrt{m}
\end{aligned}$$

where we make use of $\|\cdot\|_2 \leq \|\cdot\|_F$. Because of that,

$$\begin{aligned}
\|\mathcal{A}_2(\boldsymbol{x})\|_2 &\leq \sqrt{d} + \sqrt{dw} \leq 2\sqrt{dw}, \\
\|\mathcal{A}_3(\boldsymbol{x})\|_2 &\leq \sqrt{w}\|\mathcal{A}_2(\boldsymbol{x})\|_2 + 2w\sqrt{dw} \leq 4w\sqrt{dw}, \\
&\cdots \\
\|\mathcal{A}_\ell(\boldsymbol{x})\|_2 &\leq \sqrt{w}\|\mathcal{A}_{\ell-1}(\boldsymbol{x})\|_2 \leq 2\sqrt{dw}(2w)^{\ell-2}.
\end{aligned} \tag{17}$$

Then focus on $\mathcal{B}(\boldsymbol{x})$. Let $\mathbf{W} \in \mathbb{R}^{m \times m'}, \|\mathbf{W}\|_F \leq \sqrt{m'}, \boldsymbol{x}, \boldsymbol{x}' \in \mathbb{R}^{m'}, \boldsymbol{b} \in \mathbb{R}^m, \|\boldsymbol{b}\|_2 \leq \sqrt{m}$. Furthermore, $\|\boldsymbol{x} - \boldsymbol{x}'\|_2 \leq \epsilon$, then

$$\|\sigma(\mathbf{W}\boldsymbol{x} + \boldsymbol{b}) - \sigma(\mathbf{W}\boldsymbol{x}' + \boldsymbol{b})\|_2 \leq \|\mathbf{W}(\boldsymbol{x} - \boldsymbol{x}')\|_2 \leq \|\mathbf{W}\|_F \|\boldsymbol{x} - \boldsymbol{x}'\|_2$$

which indicates that $\|\mathcal{B}(\boldsymbol{x}) - \mathcal{B}(\boldsymbol{x})'\|_2 \leq (\sqrt{w})^{L-\ell}\|\boldsymbol{x} - \boldsymbol{x}'\|_2$

Finally, for any $\mathbf{W}, \mathbf{W}' \in \mathbb{R}^{m \times m'}, \boldsymbol{x} \in \mathbb{R}^{m'}, \boldsymbol{b}, \boldsymbol{b}' \in \mathbb{R}^m$, one have

$$\begin{aligned}
\|(\mathbf{W}\boldsymbol{x} + \boldsymbol{b}) - (\mathbf{W}'\boldsymbol{x} + \boldsymbol{b}')\|_2 &= \|(\mathbf{W} - \mathbf{W}')\boldsymbol{x} + (\boldsymbol{b} - \boldsymbol{b}')\|_2 \\
&\leq \|\mathbf{W} - \mathbf{W}'\|_2 \|\boldsymbol{x}\|_2 + \|\boldsymbol{b} - \boldsymbol{b}'\|_2. \\
&\leq \|\mathbf{W} - \mathbf{W}'\|_F \|\boldsymbol{x}\|_2 + \sqrt{m}\|\boldsymbol{b} - \boldsymbol{b}'\|_\infty.
\end{aligned}$$

In summary,

$$\begin{aligned}
|g_\ell(\boldsymbol{x}) - g'_\ell(\boldsymbol{x})| &= |\mathcal{B}_\ell(\mathbf{W}_\ell \mathcal{A}_\ell(\boldsymbol{x}) + \boldsymbol{b}_\ell) - \mathcal{B}_\ell(\mathbf{W}'_\ell \mathcal{A}_\ell(\boldsymbol{x}) + \boldsymbol{b}'_\ell)| \\
&\leq (\sqrt{w})^{L-\ell}\|(\mathbf{W}_\ell \mathcal{A}_\ell(\boldsymbol{x}) + \boldsymbol{b}_\ell) - (\mathbf{W}'_\ell \mathcal{A}_\ell(\boldsymbol{x}) + \boldsymbol{b}'_\ell)\|_2 \\
&\leq (\sqrt{w})^{L-\ell}(\|\mathbf{W}_\ell - \mathbf{W}'_\ell\|_F \|\mathcal{A}_\ell(\boldsymbol{x})\|_2 + \|\boldsymbol{b}_\ell - \boldsymbol{b}'_\ell\|_2) \\
&\leq 2^{(\ell-1)}w^{(L+\ell-3)/2}d^{1/2}\epsilon + w^{(L-\ell)/2}\bar{\epsilon}
\end{aligned}$$

Let $f(x), f'(x)$ be two neural networks satisfying (16) with parameters $W_1, b_1, \ldots, W_L, b_L$ and $W'_1, b'_1, \ldots, W'_L, b'_L$ respectively, and $\|W_\ell - W'_\ell\|_F \leq \epsilon_\ell, \|b_\ell - b'_\ell\|_F \leq \tilde{\epsilon}_\ell$. Further define $f_\ell$ be the neural network with parameters $W_1, b_1, \ldots, W_\ell, b_\ell, W'_{\ell+1}, b'_{\ell+1}, \ldots, W'_L, b'_L$, then

$$|f(x) - f'(x)| \leq |f(x) - f_1(x)| + |f_1(x) - f_2(x)| + \cdots + |f_{L-1}(x) - f'(x)|$$

$$\leq \sum_{\ell=1}^{L} 2^{(\ell-2)}d^{1/2}w^{(L+\ell-3)/2}\epsilon + w^{(L-\ell)/2}\bar{\epsilon}$$

For any $\delta > 0$, one can choose

$$\epsilon_\ell = \frac{\delta}{2^\ell w^{(L+\ell-3)/2}d^{1/2}}, \tilde{\epsilon}_\ell = \frac{\delta}{2w^{(L-\ell)/2}}$$

such that $|f(x) - f'(x)| \leq \delta$.

On the other hand, the $\epsilon$-covering number of $\{\mathbf{W} \in \mathbb{R}^{m \times m'} : \|\mathbf{W}\|_F \leq \sqrt{m'}\}$ on Frobenius norm is no larger than $(2\sqrt{m'}/\epsilon + 1)^{m \times m'}$, and the $\bar{\epsilon}$-covering number of $\{\mathbf{b} \in \mathbb{R}^m : \|\mathbf{b}\|_2 \leq 1\}$ on infinity norm is no larger than $(2/\bar{\epsilon} + 1)^m$. The entropy of this neural network can be bounded by

$$\log \mathcal{N}(f; \delta) \leq w^2 L \log(2^{L+1}w^{L-1}/\delta + 1) + wL \log(2^{L-1}w^{(L-1)/2}d^{1/2}/\delta + 1)$$

$\square$

### D.3 Covering Number of $p$-Norm Constrained Linear Combination

**Lemma 6.** *Let $\mathcal{G} \subseteq \{\mathbb{R}^d \to [-c_3, c_3]\}$ be a set with covering number satisfying $\log \mathcal{N}(\mathcal{G}, \delta) \lesssim k\log(1/\delta)$ for some finite $c_3$, and for any $g \in \mathcal{G}, |a| \leq 1$, we have $ag \in \mathcal{G}$. The covering number of $\mathcal{F} = \left\{\sum_{i=1}^M a_i g_i \Big| g_i \in \mathcal{G}, \|a\|_p^p \leq P, 0 < p < 1\right\}$ for any $P > 0$ satisfies*

$$\log \mathcal{N}(\mathcal{F}, \epsilon) \lesssim kP^{\frac{1}{1-p}}(\delta/c_3)^{-\frac{p}{1-p}}\log(c_3 P/\delta)$$

*up to a double logarithmic factor.*

*Proof.* Let $\epsilon$ be a positive constant. Without the loss of generality, we can sort the coefficients in descending order in terms of their absolute values. There exists a positive integer $\mathcal{M}$ (as a function of $\epsilon$), such that $|a_i| \geq \epsilon$ for $i \leq \mathcal{M}$, and $|a_i| < \epsilon$ for $i > \mathcal{M}$.

By definition, $\mathcal{M}\epsilon^p \leq \sum_{i=1}^{\mathcal{M}} |a_i|^p \leq P$ so $\mathcal{M} \leq P/\epsilon^p$, and $|a_i|^p \leq P, |a_i| \leq P^{1/p}$ for all $i$. Furthermore,

$$\sum_{i>m} |a_i| = \sum_{i>\mathcal{M}} |a_i|^p |a_i|^{1-p} < \sum_{i>\mathcal{M}} |a_i|^p \epsilon^{1-p} \leq P\epsilon^{1-p}$$

Let $\tilde{g}_i = \arg\min_{g \in \tilde{\mathcal{G}}} \|g - \frac{a_i}{P^{1/p}}g_i\|_\infty$ where $\tilde{\mathcal{G}}$ is the $\delta'$-convering set of $\mathcal{G}$. By definition of the covering set,

$$\left\|\sum_{i=1}^M a_i g_i(x) - \sum_{i=1}^{\mathcal{M}} P^{1/p}\tilde{g}_i(x)\right\|_\infty \leq \left\|\sum_{i=1}^{\mathcal{M}}(a_i g_i(x) - P^{1/p}\tilde{g}_i(x))\right\|_\infty + \left\|\sum_{i=\mathcal{M}+1}^M a_i g_i(x)\right\|_\infty$$
$$\leq \mathcal{M}P^{1/p}\delta' + c_3 P\epsilon^{1-p}. \tag{18}$$

Choosing

$$\epsilon = (\delta/2c_3 P)^{\frac{1}{1-p}}, \delta' \asymp P^{-\frac{1}{p(1-p)}}(\delta/2c_3)^{\frac{1}{1-p}}/2, \tag{19}$$

we have $\mathcal{M} \leq P^{\frac{1}{1-p}}(\delta/2c_3)^{-\frac{p}{1-p}}, \mathcal{M}P^{1/p}\delta' \leq \delta/2, c_3 P\epsilon^{1-p} \leq \delta/2$, so (18) $\leq \delta$. One can compute the covering number of $\mathcal{F}$ by

$$\log \mathcal{N}(\mathcal{F}, \delta) \leq \mathcal{M} \log \mathcal{N}(\mathcal{G}, \delta') \lesssim k\mathcal{M} \log(1/\delta') \tag{20}$$

Taking (19) into (20) finishes the proof. $\square$

## E Proof of Approximation Error

### E.1 Approximation of Neural Networks to B-spline Basis Functions

**Lemma 11.** *Let $M_{m,k,s}$ be the B-spline of order $m$ with scale $2^{-k}$ in each dimension and position $s \in \mathbb{R}^d$: $M_{m,k,s}(\mathbf{x}) := M_m(2^k(\mathbf{x} - s))$, $M_m$ is defined in (13). There exists a neural network with d-dimensional input and one output, with width $w_{d,m} = O(dm)$ and depth $L \lesssim \log(c_{d,m}/\epsilon)$ for some constant $c_{d,m}$ that depends only on $m$ and $d$, approximates the B spline basis function $M_{m,k,s}(\mathbf{x}) := M_m(2^k(\mathbf{x} - s))$ as defined in Section C.1. This neural network, denoted as $\tilde{M}_{m,k,s}(\mathbf{x}), \mathbf{x} \in \mathbb{R}^d$, satisfy*

- $|\tilde{M}_{m,k,\boldsymbol{s}}(\boldsymbol{x}) - M_{m,k,\boldsymbol{s}}(\boldsymbol{x})| \leq \epsilon$, *if* $0 \leq 2^k(x_i - s_i) \leq m + 1, \forall i \in [d]$,

- $\tilde{M}_{m,k,\boldsymbol{s}}(\boldsymbol{x}) = 0$, *otherwise.*

- *The weight in each layer has bounded norm* $\|\mathbf{W}^{(\ell)}\|_F \lesssim 2^{k/L}\sqrt{w}$, *except the first layer where* $\|\mathbf{W}^{(1)}\|_F \leq 2^{k/L}\sqrt{d}$.

Note that the product of the coefficients among all the layers are proportional to $2^k$, instead of $2^{km}$ when approximating truncated power basis functions. This is because the transformation from $M_m$ to $M_{m,k,\boldsymbol{s}}$ only scales the domain of the function by $2^k$, while the codomain of the function is not changed. To apply the transformation to the neural network, one only need to scale weights in the first layer by $2^k$, which is equivalent to scaling the weights in each layer bt $2^{k/L}$ and adjusting the bias according.

As for the proof, we follow the method developed in Yarotsky (2017); Suzuki (2018), while putting our attention on bounding the Frobenius norm of the weights.

**Lemma 12** (Yarotsky (2017, Proposition 3))**.** *: There exists a neural network with two-dimensional input and one output* $f_\times(x, y)$, *with constant width and depth* $O(\log(1/\delta))$, *and the weight in each layer is bounded by a global constant* $c_1$, *such that*

- $|f_\times(x, y) - xy| \leq \delta, \forall\, 0 \leq x, y \leq 1$,

- $f_\times(x, y) = 0, \forall\, x = 0$ *or* $y = 0$.

We first prove a special case of Lemma 11 on the unscaled, unshifted B-spline basis function by fixing $k = 0, \boldsymbol{s} = 0$:

**Proposition 13.** *There exists a neural network with $d$-dimensional input and one output, with width* $w = w(d, m) \asymp dm$ *and depth* $L \lesssim \log(c(m, d)/\epsilon)$ *for some constant* $w, c$ *that depends only on $m$ and $d$, denoted as* $\tilde{M}_m(\boldsymbol{x}), \boldsymbol{x} \in \mathbb{R}^d$, *such that*

- $|\tilde{M}_m(\boldsymbol{x}) - M_m(\boldsymbol{x})| \leq \epsilon$, *if* $0 \leq x_i \leq m + 1, \forall i \in [d]$, *while* $M_m(\cdot)$ *denote $m$-th order B-spline basis function,*

- $\tilde{M}_m(\boldsymbol{x}) = 0$, *if* $x_i \leq 0$ *or* $x_i \geq m + 1$ *for any* $i \in [d]$.

- *The weight in each layer has bounded norm* $\|\mathbf{W}^{(\ell)}\|_F \lesssim \sqrt{w}$.

*Proof.* We first show that one can use a neural network with constant width $w_0$, depth $L \asymp \log(m/\epsilon_1)$ and bounded norm $\|W^{(1)}\|_F \leq O(\sqrt{d}), \|W^{(\ell)}\|_F \leq O(\sqrt{w}), \forall \ell = 2, \ldots, L$ to approximate truncated power basis function up to accuracy $\epsilon_1$ in the range $[0, 1]$. Let $m = \sum_{i=0}^{\lceil \log_2 m \rceil} m_i 2^i, m_i \in \{0, 1\}$ be the binary digits of $m$, and define $\bar{m}_j = \sum_{j=0}^{i} m_j, \gamma = \lceil \log_2 m \rceil$, then for any $x$

$$
\begin{aligned}
x_+^m &= x_+^{\bar{m}_\gamma} \times \left(x_+^{2^\gamma}\right)^{m_\gamma} \\
[x_+^{\bar{m}_\gamma}, x_+^{2^\gamma}] &= [x_+^{\bar{m}_{\gamma-1}} \times \left(x_+^{2^{\gamma-1}}\right)^{m_{\gamma-1}}, x_+^{2^{\gamma-1}} \times x_+^{2^{\gamma-1}}] \\
&\quad \cdots \\
[x_+^{\bar{m}_2}, x_+^4] &= [x_+^{\bar{m}_1} \times \left(x_+^2\right)^{m_1}, x_+^2 \times x_+^2] \\
[x_+^{\bar{m}_1}, x_+^2] &= [x_+^{\bar{m}_0} \times x_+^{m_0}, x_+ \times x_+]
\end{aligned}
\tag{21}
$$

Notice that each line of equation only depends on the line immediately below. Replacing the multiply operator $\times$ with the neural network approximation shown in Lemma 12 demonstrates the architecture of such neural network approximation. For any $x, y \in [0, 1]$, let $|f_\times(x, y) - xy| \leq \delta, |x - \tilde{x}| \leq \delta_1, |y - \delta y| \leq \delta_2$, then $|f_\times(\tilde{x}, \tilde{y}) - xy| \leq \delta_1 + \delta_2 + \delta$. Taking this into (21) shows that $\epsilon_1 \asymp 2^\gamma \delta \asymp m\delta$, where $\epsilon_1$ is the upper bound on the approximate error to truncated power basis of order $m$ and $\delta$ is the approximation error to a single multiply operator as in Lemma 12.

A univariate B-spline basis can be expressed using truncated power basis, and observing that it is symmetric around $(m+1)/2$:

$$
\begin{aligned}
M_m(x) &= \frac{1}{m!} \sum_{j=1}^{m+1} (-1)^j \binom{m+1}{j} (x-j)_+^m \\
&= \frac{1}{m!} \sum_{j=1}^{\lceil (m+1)/2 \rceil} (-1)^j \binom{m+1}{j} (\min(x, m+1-x) - j)_+^m \\
&= \frac{((m+1)/2)^m}{m!} \sum_{j=1}^{\lceil (m+1)/2 \rceil} (-1)^j \binom{m+1}{j} \left( \frac{\min(x, m+1-x) - j}{(m+1)/2} \right)_+^m,
\end{aligned}
$$

A multivariate ($d$-dimensional) B-spline basis function can be expressed as the product of truncated power basis functions and thus can be decomposed as

$$
\begin{aligned}
M_m(\boldsymbol{x}) &= \prod_{i=1}^{d} M_m(x_i) \\
&= \frac{((m+1)/2)^{md}}{(m!)^d} \prod_{i=1}^{d} \left( \sum_{j=1}^{\lceil (m+1)/2 \rceil} (-1)^j \binom{m+1}{j} \left( \frac{\min(x_i, m+1-x) - j}{(m+1)/2} \right)_+^m \right) \quad (22)
\end{aligned}
$$

Using Lemma 12, one can construct $m+1$ number of neural networks, and each of them has width $w_0$ and depth $L = O(\log(m/\epsilon_1))$, such that the $(j+1)$-th neural network approximates $\left(\frac{x-j}{(m+1)/2}\right)_+^m$ with error no more than $\epsilon_1$ for any $0 \le x \le (m+1)/2$. The weighted summation of these subnetworks can approximate the univariate B-spline basis function with error no more than

$$
d((m+1)/2)^m \frac{1}{m!} \sum_{i=1}^{m+1} \binom{m+1}{j} \epsilon_1 \asymp \frac{de^{2m}}{\sqrt{m}} \epsilon_1
$$

where we applied Stirling's approximation.

A multivariate B-spline basis is the product of univariate B-spline basis along each dimension

$$
M_m(\boldsymbol{x}) = \prod_{i=1}^{d} M_m(x_i).
$$

We can construct a neural network to approximate this function by parallizing $d$ number of neural networks to approximate each B-spline basis function along each dimension, and use the last $L_1 \asymp \log(d/\delta)$ layers to approximate their product. The totol approximation error of this function is bounded by

$$
d\frac{((m+1)/2)^m}{m!} \sum_{j=1}^{m+1} \binom{m+1}{j} \epsilon_1 + (d-1)\delta \asymp \frac{e^{2m}}{\sqrt{m}} d\epsilon_1 + d\delta
$$

where $\delta$ and $\epsilon_1$ has the same definition as above. Choosing $\delta = \frac{\epsilon}{d(e^{2m}\sqrt{m}+1)}$, and recall $\epsilon_1 \asymp m\delta$ proves the approximation error.

$\square$

The proof of the Lemma 11 for general $k, \boldsymbol{s}$ follows by appending one more layer in the front, as we show below.

*Proof of Lemma 11.* Using the neural network proposed in Proposition 13, one can construct a neural network for appropximating $M_{m,k,\boldsymbol{s}}$ by adding one layer before the first layer:

$$
\sigma(2^k \mathbf{I}_d \boldsymbol{x} - 2^k \boldsymbol{s})
$$

The unused neurons in the first hidden layer is zero padded. The Frobenius norm of the weight is $2^k \|\mathbf{I}_d\|_F = 2^k \sqrt{d}$. Following the proof of Proposition 4, rescaling the weight in this layer by $2^{-k}$, and the weight matrix in the last layer by $2^k$, and scaling the bias properly, one can verify that this neural network satisfy the statement. $\qquad\square$

### E.2 SPARSE APPROXIMATION OF BESOV FUNCTIONS USING B-SPLINE WAVELETS

**Proposition 7.** *Let $\alpha - d/p > 1, r > 0$. For any function in Besov space $f_0 \in B_{p,q}^\alpha$ and any positive integer $\bar{M}$, there is an $\bar{M}$-sparse approximation using B-spline basis of order $m$ satisfying $0 < \alpha < \min(m, m - 1 + 1/p)$: $\check{f}_{\bar{M}} = \sum_{i=1}^{\bar{M}} a_{k_i, \boldsymbol{s}_i} M_{m, k_i, \boldsymbol{s}_i}$ for any positive integer $\bar{M}$ such that the approximation error is bounded as $\|\check{f}_{\bar{M}} - f_0\|_r \lesssim \bar{M}^{-\alpha/d} \|f_0\|_{B_{p,q}^\alpha}$, and the coefficients satisfy*

$$\|\{2^{k_i} a_{k_i, \boldsymbol{s}_i}\}_{k_i, \boldsymbol{s}_i}\|_p \lesssim \|f_0\|_{B_{p,q}^\alpha}.$$

**Remark 1.** *The requirement in Proposition 7: $\alpha - d/p > 1$ is stronger than the condition typically found in approximation theorem $\alpha - d/p \geq 0$ (Dũng, 2011), so-called "Boundary of continuity", or the condition in Suzuki (2018) $\alpha > d(1/p - 1/r)_+$ . This is because although the functions in $B_{p,q}^\alpha$ when $0 \leq \alpha - d/p < 1$ can be approximated by B-spline basis, the sum of weighted coefficients may not converge. One simple example is the step function $f_{step}(x) = \mathbf{1}(x \geq 0.5), f_{step} \in B_{1,\infty}^1$. Although it can be decomposed using first order B-spline basis as in (12), the summation of the coefficients is infinite. Actually one only needs a ReLU neural network with one hidden layer and two neurons to approximate this function to arbitrary precision, but the weight need to go to infinity.*

*Proof.* Dũng (2011, Theorem 3.1) Suzuki (2018, Lemma 2) proposed an adaptive sampling recovery method that approximates a function in Besov space. The method is divided into two cases: when $p \geq r$, and when $p < r$.

When $p \geq r$, there exists a sequence of scalars $\lambda_{\boldsymbol{j}}, \boldsymbol{j} \in P^d(\mu), P_d(\mu) := \{\boldsymbol{j} \in \mathbb{Z}^d : |j_i| \leq \mu, \forall i \in d\}$ for some positive $\mu$, for arbitrary positive integer $\bar{k}$, the linear operator

$$Q_{\bar{k}}(f, \boldsymbol{x}) = \sum_{\boldsymbol{s} \in J(\bar{k}, m, d)} a_{\bar{k}, \boldsymbol{s}}(f) M_{\bar{k}, \boldsymbol{s}}(\boldsymbol{x}), \quad a_{\bar{k}, \boldsymbol{s}}(f) = \sum_{\boldsymbol{j} \in \mathbb{Z}^d, P^d(\mu)} \lambda_{\boldsymbol{j}} \bar{f}(\boldsymbol{s} + 2^{-\bar{k}} \boldsymbol{j})$$

has bounded approximation error

$$\|f - Q_{\bar{k}}(f, x)\|_r \leq C 2^{-\alpha \bar{k}} \|f\|_{B_{p,q}^\alpha},$$

where $\bar{f}$ is the extrapolation of $f$, $J(\bar{k}, m, d) := \{\boldsymbol{s} : 2^{\bar{k}} \boldsymbol{s} \in \mathbb{Z}^d, -m/2 \leq 2^{\bar{k}} s_i \leq 2^{\bar{k}} + m/2, \forall i \in [d]\}$. See Dũng (2011, 2.6-2.7) for the detail of the extrapolation as well as references for options of sequence $\lambda_{\boldsymbol{j}}$.

Furthermore, $Q_{\bar{k}}(f) \in B_{p,q}^\alpha$ so it can be decomposed in the form (12) with $M = \sum_{k=0}^{\bar{k}} (2^k + m - 1)^d \lesssim 2^{\bar{k}d}$ components and $\|\{\tilde{c}_{k,\boldsymbol{s}}\}_{k,\boldsymbol{s}}\| \lesssim \|Q_{\bar{k}}(f)\|_{B_{p,q}^\alpha} \lesssim \|f\|_{B_{p,q}^\alpha}$ where $\tilde{c}_{k,\boldsymbol{s}}$ is the coefficients of the decomposition of $Q_{\bar{k}}(f)$. Choosing $\bar{k} \asymp \log_2 M/d$ leads to the desired approximation error.

On the other hand, when $p < r$, there exists a greedy algorithm that constructs

$$G(f) = Q_{\bar{k}}(f) + \sum_{k=\bar{k}+1}^{k^*} \sum_{j=1}^{n_k} c_{k, \boldsymbol{s}_j}(f) M_{k, \boldsymbol{s}_j}$$

where $\bar{k} \asymp \log_2(M), k^* = [\epsilon^{-1} \log(\lambda M)] + \bar{k} + 1, n_k = [\lambda M 2^{-\epsilon(k-\bar{k})}]$ for some $0 < \epsilon < \alpha/\delta - 1, \delta = d(1/p - 1/r), \lambda > 0$, such that

$$\|f - G(f)\|_r \leq \bar{M}^{-\alpha/d} \|f\|_{B_{p,q}^\alpha}$$

and

$$\sum_{k=0}^{\bar{k}} (2^k + m - 1)^d + \sum_{k=\bar{k}+1}^{k^*} n_k \leq \bar{M}.$$

See Dũng (2011, Theorem 3.1) for the detail.

Finally, since $\alpha - d/p > 1$,

$$
\begin{aligned}
\|\{2^{k_i} c_{k_i, \boldsymbol{s}_i}\}_{k_i, \boldsymbol{s}_i}\|_p &\leq \sum_{k=0}^{\bar{k}} 2^k \|\{c_{k_i, \boldsymbol{s}_i}\}_{\boldsymbol{s}_i}\|_p \\
&= \sum_{k=0}^{\bar{k}} 2^{(1-(\alpha-d/p))k} (2^{(\alpha-d/p)k} \|\{c_{k_i, \boldsymbol{s}_i}\}_{\boldsymbol{s}_i}\|_p) \\
&\lesssim \sum_{k=0}^{\bar{k}} 2^{(1-(\alpha-d/p))k} \|f\|_{B_{p,q}^\alpha} \\
&\approx \|f\|_{B_{p,q}^\alpha}
\end{aligned}
\tag{23}
$$

where the first line is because for arbitrary vectors $\boldsymbol{a}_i, i \in [n], \|\sum_{i=1}^n \boldsymbol{a}_i\|_p \leq \sum_{i=1}^n \|\boldsymbol{a}_i\|_p$, the third line is because the sequence norm of B-spline decomposition is equivalent to the norm in Besov space (see Section C.1) . □

Note that when $\alpha - d/p = 1$, the sequence norm (23) is bounded (up to a factor of constant) by $k^* \|f\|_{B_{p,q}^\alpha}$, which can be proven by following (23) except the last line. This adds a logarithmic term with respect to $\bar{M}$ compared with the result in Proposition 7. This will add a logarithmic factor to the MSE. We will not focus on this case in this paper of simplicity.

### E.3 SPARSE APPROXIMATION OF BESOV FUNCTIONS USING PARALLEL NEURAL NETWORKS

**Theorem 8.** *Under the same condition as Proposition 7, for any positive integer $\bar{M}$, any function in Besov space $f_0 \in B_{p,q}^\alpha$ can be approximated by a parallel neural network with no more than $O(\bar{M})$ number of subnetworks satisfying:*

1. *Each subnetwork has width $w = O(md)$ and depth $L$.*
2. *The weights in each layer satisfy $\|\bar{\mathbf{W}}_k^{(\ell)}\|_F \leq O(\sqrt{w})$ except the first layer $\|\bar{\mathbf{W}}_k^{(1)}\|_F \leq O(\sqrt{d})$,*
3. *The scaling factors have bounded $2/L$-norm: $P' := \|\{a_j\}\|_{2/L}^{2/L} \lesssim \bar{M}^{1-2/(pL)}$.*
4. *The approximation error is bounded by*
$$
\|\tilde{f} - f_0\|_r \leq (c_4 \bar{M}^{-\alpha/d} + c_5 e^{-c_6 L}) \|f\|_{B_{p,q}^\alpha}
$$
*where $c_4, c_5, c_6$ are constants that depend only on $m, d$ and $p$.*

The proof is divided into three steps:

1. Bound the 0-norm and the $p$-norm of the coefficients of B-spline basis in order to approximate an arbitrary function in Besov space up to any $\epsilon > 0$.
2. Bound $p'$-norm of the coefficients of B-spline basis functions where $p' = 2/L, 0 < p' < 1$ using the results above .
3. Add the approximation of neural network to B-spline basis computed in Lemma 11 into Step 2.

We first prove the following lemma.

**Lemma 14.** *For any $a \in \mathbb{R}^{\bar{M}}, 0 < p' < p$, it holds that:*
$$
\|a\|_{p'}^{p'} \leq \bar{M}^{1-p'/p} \|a\|_p^{p'}.
$$

*Proof.*

$$
\sum_i |a_i|^{p'} = \langle \mathbf{1}, |\boldsymbol{a}|^{p'} \rangle \leq \left( \sum_i 1 \right)^{1-\frac{p'}{p}} \left( \sum_i (|a_i|^{p'})^{\frac{p}{p'}} \right)^{\frac{p'}{p}} = \bar{M}^{1-\frac{p'}{p}} \|a\|_p^{p'}
$$

The first inequality uses a Holder's inequality with conjugate pair $\frac{p}{p'}$ and $1/(1 - \frac{p'}{p})$. $\qquad\square$

*Proof of Theorem 8.* Using Proposition 7, one can construct $\bar{M}$ number of NN according to Lemma 11, such that each NN represents one B-spline basis function. The weights in the last layer of each NN is scaled to match the coefficients in Proposition 7. Taking $p'$ in Lemma 14 as $2/L$ and combining with Lemma 11 finishes the proof. $\qquad\square$

# F  PROOF OF THE MAIN THEOREM

**Theorem 1 extended form.** *For any fixed $\alpha - d/p > 1, q \geq 1, L \geq 3$, define $m = \lceil \alpha - 1 \rceil$. For any $f_0 \in B_{p,q}^\alpha$, given an $L$-layer parallel neural network satisfying*

- *The width of each subnetwork is **fixed** satisfying $w \geq O(md)$. See Theorem 8 for the detail.*
- *The number of subnetworks is **large enough**: $M \gtrsim n^{\frac{1-2/L}{2\alpha/d+1-2/(pL)}}$.*

*Under the assumption as in Lemma 17, with proper choice of the parameter of regularizaton $\lambda$ that depends on $\mathcal{D}, \alpha, d, L$, the solution $\hat{f}$ parameterized by (2) satisfies*

$$\text{MSE}(\hat{f}) = \tilde{O}\left(\left(\frac{w^{4-4/L}L^{2-4/L}}{n^{1-2/L}}\right)^{\frac{2\alpha/d}{2\alpha/d+1-2/(pL)}} + e^{-c_6 L}\right)$$

*where $\tilde{O}$ shows the scale up to a logarithmic factor, and $c_6$ is the constant defined in Theorem 8.*

*Proof.* First recall the relationship between covering number (entropy) and estimation error:

**Proposition 15.** *Let $\mathcal{F} \subseteq \{\mathbb{R}^d \to [-F, F]\}$ be a set of functions. Assume that $\mathcal{F}$ can be decomposed into two orthogonal spaces $\mathcal{F} = \mathcal{F}_\parallel \times \mathcal{F}_\perp$ where $\mathcal{F}_\perp$ is an affine space with dimension of $N$. Let $f_0 \in \{\mathbb{R}^d \to [-F, F]\}$ be the target function and $\hat{f}$ be the least squares estimator in $\mathcal{F}$:*

$$\hat{f} = \arg\min_{f \in \mathcal{F}} \sum_{i=1}^n (y_i - f(x_i))^2, y_i = f_0(x_i) + \epsilon_i, \epsilon_i \sim \mathcal{N}(0, \sigma^2) i.i.d.,$$

*then it holds that*

$$\text{MSE}(\hat{f}) \leq \tilde{O}\left(\arg\min_{f \in \mathcal{F}} \text{MSE}(f) + \frac{N + \log \mathcal{N}(\mathcal{F}_\parallel, \delta) + 2}{n} + (F + \sigma)\delta\right).$$

The proof of Proposition 15 is defered to the section below. We choose $\mathcal{F}$ as the set of functions that can be represented by a parallel neural network as stated, the (null) space $\mathcal{F}_\perp = \{f : f(\boldsymbol{x}) = constant\}$ be the set of functions with constant output, which has dimension 1. This space captures the bias in the last layer, while the other parameters contributes to the projection in $\mathcal{F}_\parallel$. See Section D.2 for how we handle the bias in the other layers. One can find that $\mathcal{F}_\parallel$ is the set of functions that can be represented by a parallel neural network as stated, and further satisfy $\sum_{i=1}^n f(\boldsymbol{x}_i) = 0$. Because $\mathcal{F}_\parallel \subseteq \mathcal{F}, \mathcal{N}(\mathcal{F}_\parallel, \delta) \leq \mathcal{N}(\mathcal{F}, \delta)$ for all $\delta > 0$, and the latter is studied in Theorem 5.

In Theorem 1, the width of each subnetwork is no less than what is required in Theorem 8, while the depth and norm constraint are the same, so the approximation error is no more that that in Theorem 8. Choosing $r = 2, p = 2/L$, and taking Theorem 5 and Theorem 8 into this Proposition 15, one gets

$$\text{MSE}(\hat{f}) \lesssim \min_{f \in \mathcal{F}} \text{MSE}(f) + \frac{w^{2+2/(1-2/L)}L^2\sqrt{d}P'^{\frac{1}{1-2/L}}\delta^{-\frac{2/L}{1-2/L}}\log(wP'/\delta)}{n} + \delta \tag{24}$$

$$\lesssim \bar{M}^{-2\alpha/d} + \frac{w^{2+2/(1-2/L)}L^2}{n}\bar{M}^{\frac{1-2/(pL)}{1-2/L}}\delta^{-\frac{2/L}{1-2/L}}\left(\log(\bar{M}/\delta) + 3\right) + \delta,$$

where $\|f\|_{B_{p,q}^\alpha}, m$ and $d$ taken as constants. By choosing

$$\delta \approx \frac{w^{4-4/L}L^{2-4/L}\bar{M}^{1-2/(pL)}}{n^{1-2/L}}, \bar{M} \approx \left(\frac{n^{1-2/L}}{w^{4-4/L}L^{2-4/L}}\right)^{\frac{1}{2\alpha/d+1-2/(pL)}},$$

we get

$$\text{MSE}(\hat{f}) \leq \tilde{O}\left(\left(\frac{w^{4-4/L}L^{2-4/L}}{n^{1-2/L}}\right)^{\frac{2\alpha/d}{2\alpha/d+1-2/(pL)}} + e^{-c_6 L}\right) \quad (25)$$

where $\text{MSE}(\hat{f})$ shows the MSE of the solution to constrained optimization problem (7) by optimally choosing $\bar{M}$ (or $P'$).

Finally, under the assumption in Lemma 17, for any constrained optimization problem, there exists a regularized optimzation problem, whose MSE is not larger than the MSE of the constrained optimization problem up to a factor of a constant. This closes the connection between (6) and (7) and finishes the proof.

Note that the empirical risk minimizer (ERM) of the parallel nerual network satisfy that the $(2/L)$-norm of the coefficients of the parallel neural network satisfy that $\|\{a_j\}\|_{2/L}^{2/L} = \|\{\tilde{a}_{j,\bar{M}}\}\|_{2/L}^{2/L}$ where $\{\tilde{a}_{j,\bar{M}}\}$ is the coefficient of the particular $\bar{M}$-sparse approximation, although $\{a_j\}$ is not necessarily $\bar{M}$ sparse. Empirically, one only need to guarantee that during initialization, the number of sub-networks $M \geq \bar{M}$ such that the $\bar{M}$-sparse approximation is feasible, thus the approximation error bound from Theorem 8 can be applied. Theorem 8 also says that $\|\{a_j\}\|_{2/L}^{2/L} = \|\{\tilde{a}_{j,\bar{M}}\}\|_{2/L}^{2/L} \lesssim \bar{M}^{1-2/pL}$, thus we can apply the covering number bound from Theorem 5 with $P' = \bar{M}^{1-2/pL}$. Finally, if $\lambda$ is optimally chosen, then it achieves a smaller MSE than this particular $\lambda'$, which has been proven to be no more than $O(\bar{M}^{-\alpha/d})$ and completes the proof.

$\square$

*Proof of Proposition 15.* For any function $f \in \mathcal{F}$, define $f_\perp = \arg\min_{h \in \mathcal{F}_\perp} \sum_{i=1}^n (f(\boldsymbol{x}_i) - h(\boldsymbol{x}_i))^2$ be the projection of $f$ to $\mathcal{F}_\perp$, and define $f_\parallel = f - f_\perp$ be the projection to the orthogonal complement. Note that $f_\parallel$ is not necessarily in $\mathcal{F}_\parallel$. However, if $f \in \mathcal{F}$, then $f_\parallel \in \mathcal{F}_\parallel$. $y_{i\perp}$ and $y_{i\parallel}$ are defined by creating a function $f_y$ such that $f_y(\boldsymbol{x}_i) = y_i, \forall i$, e.g. via interpolation. Because $\mathcal{F}_\parallel$ and $\mathcal{F}_\perp$ are orthogonal, the empirical loss and population loss can be decomposed in the same way:

$$L_\parallel(f) = \frac{1}{n}\sum_{i=1}^n (f_\parallel(\boldsymbol{x}) - f_{0\parallel}(\boldsymbol{x}))^2 + \frac{n-N}{n}\sigma^2, \qquad L_\perp(f) = \frac{1}{n}\sum_{i=1}^n (f_\perp(\boldsymbol{x}) - f_{0\perp}(\boldsymbol{x}))^2 + \frac{N}{n}\sigma^2,$$

$$\hat{L}_\parallel(f) = \frac{1}{n}\sum_{i=1}^n (f_\parallel(\boldsymbol{x}) - y_{i\parallel})^2, \qquad \hat{L}_\perp(f) = \frac{1}{n}\sum_{i=1}^n (f_\perp(\boldsymbol{x}) - y_{i\perp}(\boldsymbol{x}))^2,$$

$$MSE_\parallel(f) = \mathbb{E}_\mathcal{D}\left[\frac{1}{n}\sum_{i=1}^n (f_\parallel(\boldsymbol{x}) - f_{0\parallel}(\boldsymbol{x}))^2\right], \qquad MSE_\perp(f) = \mathbb{E}_\mathcal{D}\left[\frac{1}{n}\sum_{i=1}^n (f_\perp(\boldsymbol{x}) - f_{0\perp}(\boldsymbol{x}))^2\right],$$

such that $L(f) = L_\parallel(f) + L_\perp(f), \hat{L}(f) = \hat{L}_\parallel(f) + \hat{L}_\perp(f)$. This can be verified by decomposing $\hat{f}, f_0$ and $y$ into two orthogonal components as shown above, and observing that $\sum_{i=1}^n f_{1\perp}(\boldsymbol{x}_i)f_{2\parallel}(\boldsymbol{x}_i) = 0, \forall f_1, f_2$.

**First prove the following claim**

**Claim 16.** *Assume that $\hat{f} = \arg\min_{f \in \mathcal{F}} \hat{L}(f)$ is the empirical risk minimizer. Then $\hat{f}_\perp = \arg\min_{f \in \mathcal{F}_\perp} \hat{L}_\perp(f), \hat{f}_\parallel = \arg\min_{f \in \mathcal{F}_\parallel} \hat{L}_\parallel(f)$, where $\hat{f}_\perp$ is the projections of $\hat{f}$ in $\mathcal{F}_\perp$, and $\hat{f}_\parallel = \hat{f} - \hat{f}_\perp$ respectively.*

*Proof.* Since $\hat{f} \in \mathcal{F}$, by definition $\hat{f}_\parallel \in \mathcal{F}_\parallel$. Assume that there exist $\hat{f}'_\perp, \hat{f}'_\parallel$, and either $\hat{L}_\perp(\hat{f}'_\perp) < \hat{L}_\perp(\hat{f}_\perp)$, or $\hat{L}_\parallel(\hat{f}'_\parallel) < \hat{L}_\parallel(\hat{f}_\parallel)$. Then

$$\hat{L}(\hat{f}') = \hat{L}(\hat{f}'_\perp + \hat{f}'_\parallel) = \hat{L}_\parallel(\hat{f}'_\perp + \hat{f}'_\parallel) + \hat{L}_\perp(\hat{f}'_\perp + \hat{f}'_\parallel) = \hat{L}_\parallel(\hat{f}'_\parallel) + \hat{L}_\perp(\hat{f}'_\perp)$$
$$< \hat{L}_\parallel(\hat{f}_\parallel) + \hat{L}_\perp(\hat{f}_\perp) = \hat{L}_\parallel(\hat{f}_\perp + \hat{f}_\parallel) + \hat{L}_\perp(\hat{f}_\perp + \hat{f}_\parallel) = \hat{L}(\hat{f})$$

which shows that $\hat{f}$ is not the minimizer of $\hat{L}(f)$ and violates the assumption.

$$\square$$

**Then we bound** $MSE_\perp(f)$**.** We convert this part into a finite dimension least square problem:

$$
\begin{aligned}
\hat{f}_\perp &= \arg\min_{f \in \mathcal{F}_\perp} \hat{L}_\perp(f) \\
&= \arg\min_{f \in \mathcal{F}_\perp} \frac{1}{n} \sum_{i=1}^n (f(\boldsymbol{x}_i) - f_{0\perp}(\boldsymbol{x}_i) - \epsilon_{i\perp})^2 \\
&= \arg\min_{f \in \mathcal{F}_\perp} \frac{1}{n} \sum_{i=1}^n (f(\boldsymbol{x}_i) - f_{0\perp}(\boldsymbol{x}_i) - \epsilon_{i\perp})^2 + \epsilon_{i\|}^2 \\
&= \arg\min_{f \in \mathcal{F}_\perp} \frac{1}{n} \sum_{i=1}^n (f(\boldsymbol{x}_i) - f_{0\perp}(\boldsymbol{x}_i) - \epsilon_{i\perp} - \epsilon_{i\|})^2 \\
&= \arg\min_{f \in \mathcal{F}_\perp} \frac{1}{n} \sum_{i=1}^n (f(\boldsymbol{x}_i) - f_{0\perp}(\boldsymbol{x}_i) - \epsilon_i)^2
\end{aligned}
$$

The forth line comes from our assumption that $\mathcal{F}_\perp$ is orthogonal to $\mathcal{F}_\|$, so $\forall f \in \mathcal{F}_\perp, f + f_{0\perp} + \epsilon_\perp$ is orthogonal to $\epsilon_\|$.

Let the basis function of $\mathcal{F}_\perp$ be $h_1, h_2, \ldots, h_N$, the above problem can be reparameterized as

$$
\arg\min_{\boldsymbol{\theta} \in \mathbb{R}^N} \frac{1}{n} \|\mathbf{X}\boldsymbol{\theta} - \boldsymbol{y}\|^2
$$

where $\mathbf{X} \in \mathbb{R}^{n \times N} : X_i = h_j(\boldsymbol{x}_i), \boldsymbol{y} = \boldsymbol{y}_{0\perp} + \boldsymbol{\epsilon}, \boldsymbol{y}_{0\perp} = [f_{0\perp}(x_1), \ldots, f_{0\perp}(x_n)], \boldsymbol{\epsilon} = [\epsilon_1, \ldots, \epsilon_n]$. This problem has a closed-form solution

$$
\boldsymbol{\theta} = (\mathbf{X}^T\mathbf{X})^{-1}\mathbf{X}^T\boldsymbol{y}
$$

Observe that $f_{0\perp} \in \mathcal{F}_\perp$, let $\boldsymbol{y}_{0\perp} = \mathbf{X}\boldsymbol{\theta}^*$, The MSE of this problem can be computed by

$$
\begin{aligned}
L(\hat{f}_\perp) &= \frac{1}{n}\|\mathbf{X}\boldsymbol{\theta} - \boldsymbol{y}_{0\perp}\|^2 = \frac{1}{n}\|\mathbf{X}(\mathbf{X}^T\mathbf{X})^{-1}\mathbf{X}^T(\mathbf{X}\boldsymbol{\theta}^* + \boldsymbol{\epsilon}) - \mathbf{X}\boldsymbol{\theta}^*\|^2 \\
&= \frac{1}{n}\|\mathbf{X}(\mathbf{X}^T\mathbf{X})^{-1}\mathbf{X}^T\boldsymbol{\epsilon}\|^2
\end{aligned}
$$

Observing that $\Pi := \mathbf{X}(\mathbf{X}^T\mathbf{X})^{-1}\mathbf{X}^T$ is an idempotent and independent projection whose rank is $N$, and that $\mathbb{E}[\boldsymbol{\epsilon}\boldsymbol{\epsilon}^T] = \sigma^2\mathbf{I}$, we get

$$
\mathrm{MSE}_\perp(\hat{f}_\perp) = \mathbb{E}[L(\hat{f}_\perp)] = \frac{1}{n}\|\Pi\boldsymbol{\epsilon}\|^2 = \frac{1}{n}\mathrm{tr}(\Pi\boldsymbol{\epsilon}\boldsymbol{\epsilon}^T) = \frac{\sigma^2}{n}\mathrm{tr}(\Pi)
$$

which concludes that

$$
\mathrm{MSE}_\perp(\hat{f}) = O\left(\frac{N}{n}\sigma^2\right). \tag{26}
$$

See also (Hsu et al., 2011, Proposition 1).

**Next we study** $\mathrm{MSE}_\|(\hat{f})$**.** Denote $\tilde{\sigma}_\|^2 = \frac{1}{n}\sum_{i=1}^n \epsilon_{i\|}^2, E = \max_i |\epsilon_i|$. Using Jensen's inequality and union bound, we have

$$
\exp(t\mathbb{E}[E]) \le \mathbb{E}[\exp(tE)] = \mathbb{E}[\max \exp(t|\epsilon_i|)] \le \sum_{i=1}^n \mathbb{E}[\exp(t|\epsilon_i|)] \le 2n\exp(t^2\sigma^2/2)
$$

Taking expectation over both sides, we get

$$
\mathbb{E}[E] \le \frac{\log 2n}{t} + \frac{t\sigma^2}{2}
$$

maximizing the right hand side over $t$ yields

$$
\mathbb{E}[E] \le \sigma\sqrt{2\log 2n}.
$$

Let $\tilde{\mathcal{F}}_{\|}$ be the covering set of $\mathcal{F}_{\|} = \{f_{\|} : f \in \mathcal{F}\}$. For any $\tilde{f}_{\|} \in \tilde{\mathcal{F}}_{\|}$,

$$
\begin{aligned}
L_{\|}(f_j) - \hat{L}_{\|}(f_j) &= \frac{1}{n}\sum_{i=1}^{n}(f_{j\|}(\boldsymbol{x}_i) - f_{0\|}(\boldsymbol{x}_i))^2 - \frac{1}{n}\sum_{i=1}^{n}(\tilde{f}_{\|}(\boldsymbol{x}_i) - y_{i\|})^2 + \frac{n-N}{n}\sigma^2 \\
&= \frac{1}{n}\sum_{i=1}^{n}\epsilon_{i\|}(2\tilde{f}_{\|}(\boldsymbol{x}_i) - f_{0\|}(\boldsymbol{x}_i) - y_{i\|}) + \frac{n-N}{n}\sigma^2 \\
&= \frac{1}{n}\sum_{i=1}^{n}\epsilon_i(2\tilde{f}_{\|}(\boldsymbol{x}_i) - f_{0\|}(\boldsymbol{x}_i) - y_{i\|}) + \frac{n-N}{n}\sigma^2 \\
&= \frac{1}{n}\sum_{i=1}^{n}\epsilon_i(2\tilde{f}_{\|}(\boldsymbol{x}_i) - 2f_{0\|}(\boldsymbol{x}_i)) + \frac{n-N}{n}\sigma^2 - \tilde{\sigma}_{\|}^2
\end{aligned}
$$

The first term can be bounded using Bernstein's inequality: let $h_i = \epsilon_i(f_{j\|}(\boldsymbol{x}_i) - f_{0\|}(\boldsymbol{x}_i))$, by definition $|h_i| \leq 2EF$,

$$
\begin{aligned}
\mathrm{Var}[h_i] &= \mathbb{E}[\epsilon_i^2(\tilde{f}_{\|}(\boldsymbol{x}_i) - f_{0\|}(\boldsymbol{x}_i))^2] \\
&= (\tilde{f}_{\|}(\boldsymbol{x}_i) - f_{0\|}(\boldsymbol{x}_i))^2 \mathbb{E}[\epsilon_i^2] \\
&= (\tilde{f}_{\|}(\boldsymbol{x}_i) - f_{0\|}(\boldsymbol{x}_i))^2 \sigma^2
\end{aligned}
$$

using Bernstein's inequality, for any $\tilde{f}_{\|} \in \tilde{\mathcal{F}}_{\|}$, with probably at least $1 - \delta_p$,

$$
\begin{aligned}
\frac{1}{n}\sum_{i=1}^{n}\epsilon_i(2\tilde{f}_{\|}(\boldsymbol{x}_i) - 2f_{0\|}(\boldsymbol{x}_i)) &= \frac{2}{n}\sum_{i=1}^{n}h_i \\
&\leq \frac{2}{n}\sqrt{2\sum_{i=1}^{n}\left(\tilde{f}_{\|}(\boldsymbol{x}_i) - f_{0\|}(\boldsymbol{x}_i)\right)^2\sigma^2\log(1/\delta_p)} + \frac{8EF\log(1/\delta_p)}{3n} \\
&= 2\sqrt{\left(L_{\|}(\tilde{f}_{\|}) - \frac{n-N}{n}\sigma^2\right)\frac{2\sigma^2\log(1/\delta_p)}{n}} + \frac{8EF\log(1/\delta_p)}{3n} \\
&\leq \epsilon\left(L_{\|}(\tilde{f}_{\|}) - \frac{n-N}{n}\sigma^2\right) + \frac{8\sigma^2\log(1/\delta_p)}{n\epsilon} + \frac{8EF\log(1/\delta_p)}{3n}
\end{aligned}
$$

the last inequality holds true for all $\epsilon > 0$. The union bound shows that with probably at least $1 - \delta$, for all $\tilde{f}_{\|} \in \tilde{\mathcal{F}}_{\|}$,

$$
\begin{aligned}
L_{\|}(\tilde{f}_{\|}) - \hat{L}_{\|}(\tilde{f}_{\|}) &\leq \epsilon\left(L_{\|}(\tilde{f}_{\|}) - \frac{n-N}{n}\sigma^2\right) + \frac{8\sigma^2\log(\mathcal{N}(\mathcal{F}_{\|},\delta)/\delta_p)}{n\epsilon} + \frac{8EF\log(\mathcal{N}(\mathcal{F}_{\|},\delta)/\delta_p)}{3n} \\
&\quad + \frac{n-N}{n}\sigma^2 - \tilde{\sigma}_{\|}^2.
\end{aligned}
$$

By rearranging the terms and using the definition of $L(\tilde{f}_{\|})$, we get

$$
(1-\epsilon)\left(L_{\|}(\tilde{f}_{\|}) - \frac{n-N}{n}\sigma^2\right) \leq \hat{L}_{\|}(\tilde{f}_{\|}) + \frac{8\sigma^2\log(\mathcal{N}(\mathcal{F}_{\|},\delta)/\delta_p)}{n\epsilon} + \frac{8EF\log(\mathcal{N}(\mathcal{F}_{\|},\delta)/\delta_p)}{3n} - \tilde{\sigma}_{\|}^2.
$$

Taking the expectation (over $\mathcal{D}$) on both sides, and notice that $\mathbb{E}[\tilde{\sigma}_{\|}^2] = \frac{n-N}{n}\sigma^2$. Furthermore, for any random variable $X, \mathbb{E}[X] = \int_{-\infty}^{\infty} x dP(X \leq x)$, we get

$$
\begin{aligned}
\max_{\tilde{f}_{\|} \in \tilde{\mathcal{F}}_{\|}} &\left((1-\epsilon)\mathrm{MSE}_{\|}(\tilde{f}_{\|}) - \mathbb{E}[\hat{L}_{\|}(\tilde{f}_{\|})]\right) \\
&\leq \left(\frac{8\sigma^2}{n\epsilon} + \frac{8F\sigma\sqrt{2\log 2n}}{3n}\right)\left(\log\mathcal{N}(\mathcal{F}_{\|},\delta) - \int_{\delta=0}^{1}\log(\delta_p)d\delta_p\right) - \frac{n-N}{n}\sigma^2 \qquad (27) \\
&= \left(\frac{8\sigma^2}{n\epsilon} + \frac{8F\sigma\sqrt{2\log 2n}}{3n}\right)(\log\mathcal{N}(\mathcal{F}_{\|},\delta) + 1) - \frac{n-N}{n}\sigma^2.
\end{aligned}
$$

where the integration can be computed by replacing $\delta$ with $e^x$. Though it is not integrable under Riemann integral, it is integrable under Lebesgue integration.

Similarly, let $\check{f}_\| = \arg\min_{f \in \mathcal{F}_\|} L_\|(f)$,

$$L_\|(\check{f}_\|) - \hat{L}_\|(\check{f}_\|) = \frac{1}{n}\sum_{i=1}^n \epsilon_i(2\check{f}_\|(\boldsymbol{x}_i) - 2f_{0\|}(\boldsymbol{x}_i)) + \frac{n-N}{n}\sigma^2 - \tilde{\sigma}_\|^2$$

with probably at least $1 - \delta_q$, for any $\epsilon > 0$,

$$-\frac{1}{n}\sum_{i=1}^n \epsilon_i(2\check{f}_\|(\boldsymbol{x}_i) - 2f_{0\|}(\boldsymbol{x}_i)) \le \epsilon\Big(L_\|(\check{f}_\|) - \frac{n-N}{n}\sigma^2\Big) + \frac{8\sigma^2\log(1/\delta_p)}{n\epsilon} + \frac{8EF\log(1/\delta_p)}{3n},$$

$$\hat{L}_\|(\check{f}_\|) \le (1+\epsilon)\Big(L_\|(\check{f}_\|) - \frac{n-N}{n}\sigma^2\Big) + \frac{8\sigma^2\log(1/\delta_p)}{n\epsilon} + \frac{8EF\log(1/\delta_q)}{3n} + \tilde{\sigma}_\|^2.$$

Taking the expectation on both sides,

$$\mathbb{E}[\hat{L}_\|(\check{f}_\|)] \le (1+\epsilon)\mathrm{MSE}_\|(\check{f}_\|) + \frac{8\sigma^2}{n\epsilon} + \frac{8F\sigma\sqrt{2\log 2n}}{3n} + \frac{n-N}{n}\sigma^2. \tag{28}$$

Finally, let $\hat{f}_* := \arg\min_{f \in \tilde{\mathcal{F}}_\|} \sum_{i=1}^n (\hat{f}_\|(\boldsymbol{x}_i) - f(\boldsymbol{x}_i))^2$ be the projection of $\hat{f}_\|$ in its $\delta$-covering space,

$$\begin{aligned}
\mathrm{MSE}_\|(\hat{f}_\|) &= \mathbb{E}\Big[\frac{1}{n}\sum_{i=1}^n (\hat{f}_\|(\boldsymbol{x}_i) - f_{0\|}(\boldsymbol{x}_i))^2\Big] \\
&= \mathbb{E}\Big[\frac{1}{n}\sum_{i=1}^n (\hat{f}_*(\boldsymbol{x}_i) - f_{0\|}(\boldsymbol{x}_i))^2 + \frac{1}{n}\sum_{i=1}^n (\hat{f}_\|(\boldsymbol{x}_i) - \hat{f}_*(\boldsymbol{x}_i))(\hat{f}_\|(\boldsymbol{x}_i) + \hat{f}_*(\boldsymbol{x}_i) - 2f_{0\|}(\boldsymbol{x}_i))\Big] \\
&\le \mathbb{E}\Big[\frac{1}{n}\sum_{i=1}^n (\hat{f}_*(\boldsymbol{x}_i) - f_{0\|}(\boldsymbol{x}_i))^2\Big] + 4F\delta \\
&= \mathrm{MSE}_\|(\hat{f}_*(\boldsymbol{x}_i)) + 4F\delta,
\end{aligned}$$

and similarly

$$\hat{L}_\|(\hat{f}_*) \le \hat{L}_\|(\hat{f}_\|) + (4F + 2E)\delta. \tag{29}$$

We can conclude that

$$\begin{aligned}
\mathrm{MSE}_\|(\hat{f}_\|) &\le \frac{1}{1-\epsilon}\Big(\mathbb{E}[\hat{L}_\|(\hat{f}_*)] + \Big(\frac{8\sigma^2}{n\epsilon} + \frac{8F\sigma\sqrt{2\log 2n}}{3n}\Big)(\log\mathcal{N}(\mathcal{F}_\|,\delta) + 1) - \frac{n-N}{n}\sigma^2\Big) \\
&\quad + 4F\delta \\
&\le \frac{1}{1-\epsilon}\Big(\mathbb{E}[\hat{L}_\|(\hat{f}_\|)] + (4F + \sigma\sqrt{8\log 2n})\delta \\
&\quad + \Big(\frac{8\sigma^2}{n\epsilon} + \frac{8F\sigma\sqrt{2\log 2n}}{3n}\Big)(\log\mathcal{N}(\mathcal{F}_\|,\delta) + 1) - \frac{n-N}{n}\sigma^2\Big) + 4F\delta \\
&\le \frac{1}{1-\epsilon}\Big(\mathbb{E}[\hat{L}_\|(\check{f}_\|)] + (4F + \sigma\sqrt{8\log 2n})\delta \\
&\quad + \Big(\frac{8\sigma^2}{n\epsilon} + \frac{8F\sigma\sqrt{2\log 2n}}{3n}\Big)(\log\mathcal{N}(\mathcal{F}_\|,\delta) + 1) - \frac{n-N}{n}\sigma^2\Big) + 4F\delta \\
&\le \frac{1+\epsilon}{1-\epsilon}\mathrm{MSE}_\|(\check{f}_\|) + \frac{1}{n}\Big(\frac{8\sigma^2}{\epsilon} + \frac{8F\sigma\sqrt{2\log 2n}}{3}\Big)\Big(\frac{\log\mathcal{N}(\mathcal{F}_\|,\delta) + 2}{1-\epsilon}\Big) \\
&\quad + \Big(4F + \frac{4F + \sigma\sqrt{8\log 2n}}{1-\epsilon}\Big)\delta,
\end{aligned}$$

where the first line comes from (27), and second comes from (29), the thid line is because $\hat{f}_\| = \arg\min_{f \in \mathcal{F}_\|} \hat{L}_\|(f)$, and the last line comes from (28). We also use that fact that $\hat{L}_\|(\hat{f}) \le \hat{L}_\|(f), \forall f$. Noticing that $\mathrm{MSE}(\hat{f}) = \mathrm{MSE}_\|(\hat{f}) + \mathrm{MSE}_\perp(\hat{f})$, combining this with (26) finishes the proof. $\square$

**Lemma 17.** *Assume that these exists $C_1, C_2 > 1$ (which may depend on the target function), for all $P' > 0$, there exists $\lambda > 0$, such that the soltion to the regularized optimization problem (6), denoted as $\tilde{f}$, satisfy*

$$C_1 P' \leq \|\{\tilde{a}_j\}\|_{2/L}^{2/L} \leq C_2 P',$$

*then the MSE of the regularized optimization problem satisfy*

$$\mathrm{MSE}(\tilde{f}) \leq C \mathrm{MSE}(\hat{f})$$

*where $C$ is a constant that depends on $C_1, C_2$, $\hat{f}$ is the solution to the constrained optimzation problem (7), and*

$$\lambda \lesssim \frac{\mathrm{MSE}(\hat{f})}{P'} \lesssim n^{-(1-2/L)}$$

*Proof.* The MSE of the regularized problem can be achieved by taking our assumtion into (4). We only need to prove the selection of $\lambda$. We apply the decomposition as in Proposition 15, and only need to consider $\mathcal{F}_\|$, as $\mathcal{F}_\perp$ is not imfluenced by regularization or constrained. From the definition of $\tilde{f}$ and $\lambda$, we have

$$\hat{L}(\tilde{f}) + \lambda\|\{\tilde{a}_j\}\|_{2/L}^{2/L} \leq \hat{L}(\hat{f}) + \lambda\|\{\hat{a}_j\}\|_{2/L}^{2/L},$$

$$\hat{L}_\|(\tilde{f}) + \lambda\|\{\tilde{a}_j\}\|_{2/L}^{2/L} \leq L_\|(\hat{f}) + \lambda\|\{\hat{a}_j\}\|_{2/L}^{2/L}$$

From Proposition 15, we get

$$(1 - \epsilon)\mathrm{MSE}(\tilde{f}) - O\left(\frac{\log\mathcal{N}(\|\{\tilde{a}_j\}\|_{2/L}^{2/L}, \delta)}{n}\right) + \lambda\|\{\tilde{a}_j\}\|_{2/L}^{2/L}$$

$$\leq (1 + \epsilon)\mathrm{MSE}(\hat{f}) + O\left(\frac{\log\mathcal{N}(\|\{\hat{a}_j\}\|_{2/L}^{2/L}, \delta)}{n}\right) + \lambda\|\{\hat{a}_j\}\|_{2/L}^{2/L} \tag{30}$$

Observing that $\mathrm{MSE}(\tilde{f}) \geq 0$, and $\frac{\log\mathcal{N}(\|\{\hat{a}_j\}\|_{2/L}^{2/L}, \delta)}{n} \approx MSE(\hat{f})$ for the optimally chosen $P'$, taking the assumtion into the inequality proves the choice of $\lambda$. $\square$

**Remark 2.** *Define $R(\lambda) := R(\arg\min \hat{L}(f) + \lambda R(f))$, where $R(f) = \|\{a_j\}\|_{2/L}^{2/L}$ is the regularizer term of a parallel NN ($f$). Notice that $R(\lambda)$ is a non-increasing function of $lambda$ (as proved below), the assumption in Lemma 17 is equivalent to that if $R(\lambda)$ contains any uncontinuous points, then the uncontinuous points should not be larger than $\frac{C_2}{C_1}$ in ratio. On the other hand, if $\lambda$ is chosen as $\lambda = O(\frac{\mathrm{MSE}(\hat{f})}{P'})$, then from (30), we get*

$$\lambda\|\{\tilde{a}_j\}\|_{2/L}^{2/L} \leq O(\mathrm{MSE}(\hat{f})) + O\left(\frac{\log\mathcal{N}(\|\{\hat{a}_j\}\|_{2/L}^{2/L}, \delta)}{n}\right)$$

$$\leq O(\mathrm{MSE}(\hat{f})) + \frac{1}{n}\tilde{O}((\|\{\hat{a}_j\}\|_{2/L}^{2/L})^{\frac{1}{1-2/L}})$$

*If the constant term in $\lambda$ is large enough, the above inequality yields two sets of solutions:*

$$\|\{\tilde{a}_j\}\|_{2/L}^{2/L} \leq O\left(\|\{\hat{a}_j\}\|_{2/L}^{2/L} + \frac{1}{\lambda}\mathrm{MSE}(\hat{f})\right) = O(\|\{\hat{a}_j\}\|_{2/L}^{2/L}),$$

*and*

$$\|\{\tilde{a}_j\}\|_{2/L}^{2/L} \geq \tilde{O}\left((n\lambda)^{\frac{1-2/L}{2/L}}\right).$$

*In the first case, one can easily see from (30) that $\mathrm{MSE}(\tilde{f}) \leq O(\mathrm{MSE}(\hat{f}))$, which says that the MSE of the regularized problem is close to the minimax rate; in the later case, the generalization gap of the regularized problem is bounded by $O(n^{\frac{1-2/L}{2/L}}\lambda^{L/2})$, which is much larger than the former case. So a sufficient condition of the above assumption is that the model does not overfit significantly (by*

*orders of magnitude) more than the constrained version. In our experiment, we find that the latter case is very difficult to happen, possibly because of the implicit regularization during training, and the connection between $\lambda$ and effective degree of freedom is actually smooth. Notably, as $L$ gets larger, in the second case $\|\{\tilde{a}_j\}\|_{2/L}^{2/L}$ increases exponentially with $L$ (the constant terms depends at most polynomially on $L$), which suggests that the latter case is less likely to happen for deep neural networks.*

**Claim 18.** *For fixed $\mathcal{D}$, the regularized problem satisfy that $R(\lambda)$ as defined above is strictly non-increasing with $\lambda$.*

*Proof.* We provide a short proof by contradiction: suppose that there exists $lambda_1 < \lambda_2$, and the solution satisfy $(f_1) < R(f_2)$ where $R(f) = \|\{a_j\}\|_{2/L}^{2/L}$ is the regularizer term of a parallel NN, $f_1, f_2$ are the solution to the regularized problem with $\lambda = \lambda_1, \lambda_2$ respectively. Then by definition of $f_1, f_2$, we have $\hat{L}(f_1) + \lambda_1 R(f_1) \le \hat{L}(f_2) + \lambda_1 R(f_2)$, so $\lambda_1 \ge \frac{\hat{L}(f_1) - \hat{L}(f_2)}{R(f_2) - R(f_1)}$; $\hat{L}(f_2) + \lambda_2 R(f_2) \le \hat{L}(f_1) + \lambda_2 R(f_1)$, so $\lambda_2 \le \frac{\hat{L}(f_1) - \hat{L}(f_2)}{R(f_2) - R(f_1)}$ which is controversal to our assumption that $\lambda_1 < \lambda_2$. $\square$

## G   MORE DISCUSSION ABOUT THE MAIN RESULT

**Representation learning and adaptivity.** The results also shed a light on the role of representation learning in DNN's ability to adapt. Specifically, different from the two-layer NN in (Parhi & Nowak, 2021a), which achieves the minimax rate of $BV(m)$ by choosing appropriate activation functions using each $m$, each subnetwork of a parallel NN can learn to approximate the spline basis of an arbitrary order, which means that if we choose $L$ to be sufficiently large, such Parallel NN with optimally tuned $\lambda$ is simultaneously near optimal for $m = 1, 2, 3, \ldots$. In fact, even if different regions of the space has different *orders* of smoothness, the paralle NN will still be able to learn appropriate basis functions in each local region. To the best of our knowledge, this is a property that none of the classical nonparametric regression methods possess.

**Synthesis v.s. analysis methods.** Our result could also inspire new ideas in estimator design. There are two families of methods in non-parametric estimation. One called *synthesis* framework which focuses on constructing appropriate basis functions to encode the contemplated structures and regress the data to such basis, e.g., wavelets (Donoho et al., 1998). The other is called *analysis* framework which uses analysis regularization on the data directly (see, e.g., RKHS methods (Scholkopf & Smola, 2001) or trend filtering (Tibshirani, 2014)). It appears to us that parallel NN is doing both simultaneously. It has a parametric family capable to synthesizing an $O(n)$ subset of an exponentially large family of basis, then *implicitly* use sparsity-inducing analysis regularization to select the relevant basis functions. In this way the estimator does not actually have to explicitly represent that exponentially large set of basis functions, thus computationally more efficient.

**Random design problem.** This paper focuses on the fixed design problem such that the results are comparable to that in nonparametric regression. One can easily apply the technique in this paper to achieve the estimation error bound on the random design problem:

**Theorem 19.** *Under the same condition as Theorem 1, the solution $\hat{f}$ parameterized by (2) satisfies*

$$\mathbb{E}_{\mathcal{D}}\mathbb{E}_f \mathrm{MSE}(\hat{f}) \le \tilde{O}\left( \left(\frac{w^{4-4/L} L^{2-4/L}}{n^{1-2/L}}\right)^{\frac{2\alpha/d}{2\alpha/d + 1 - 2/(pL)}} + e^{-c_6 L} \right)$$

*where $\tilde{O}$ shows the scale up to a logarithmic factor, and $c_6$ is the constant defined in Theorem 8, $\mathbb{E}_{\mathcal{D}}$ indicates that the expectation is taken with respect to the training set $\mathcal{D}$, $\mathbb{E}_f$ indicates that the expectation is taken with respect to the domain of $\hat{f}$.*

The proof is similar to that of Theorem 1. The main difference lays in the proof of the estimation error. For $f_\perp$ part, the estimation error can be bounded using VC-dimension, which is 1. For $f_\parallel$ part, the estimation error can be bounded using its covering number, e.g. Lemma 8 in Schmidt-Hieber (2020).

# H DETAILED EXPERIMENTAL SETUP

## H.1 TARGET FUNCTIONS

The doppler function used in Figure 3(d)-(f) is

$$f(x) = \sin(4/(x + 0.01)) + 1.5.$$

The "vary" function used in Figure 3(g)-(i) is

$$\begin{aligned}
f(x) = &M_1(x/0.01) + M_1((x - 0.02)/0.02) + M_1((x - 0.06)/0.03) \\
&+ M_1((x - 0.12)/0.04) + M_3((x - 0.2)/0.02) + M_3((x - 0.28)/0.04) \\
&+ M_3((x - 0.44)/0.06) + M_3((x - 0.68)/0.08),
\end{aligned}$$

where $M_1, M_3$ are first and third order Cardinal B-spline bases functions respectively. We uniformly take 256 samples from 0 to 1 in the piecewise cubic function experiment, and uniformly 1000 samples from 0 to 1 in the doppler function and "vary" function experiment. We add zero mean independent (white) Gaussian noise to the observations. The standard derivation of noise is 0.4 in the doppler function experiment and 0.1 in the "vary" function experiment.

## H.2 TRAINING/FITTING METHOD

In the piecewise polynomial function ("vary") experiment, the depth of the PNN $L = 10$, the width of each subnetwork $w = 10$, and the model contains $M = 500$ subnetworks. The depth of NN is also 10, and the width is 240 such that the NN and PNN have almost the same number of parameters. In the doppler function experiment, the depth of the PNN $L = 12$, the width of each subnetwork $w = 10$, and the model contains $M = 2000$ subnetworks, because this problem requires a more complex model to fit. The depth of NN is 12, and the width is 470. We used Adam optimizer with learning rate of $10^{-3}$. We first train the neural network layer by layer without weight decay. Specifically, we start with a two-layer neural network with the same number of subnetworks and the same width in each subnetwork, then train a three layer neural network by initializing the first layer using the trained two layer one, until the desired depth is reached. After that, we turn the weight decay parameter and train it until convergence. In both trend filtering and smoothing spline experiment, the order is 3, and in wavelet denoising experiment, we use sym4 wavelet with soft thresholding. We implement the trend filtering problem according to Tibshirani (2014) using CVXPY, and use MOSEK to solve the convex optimization problem. We directly call R function $smooth.spline$ to solve smoothing spline.

## H.3 POST PROCESSING

The degree of freedom of smoothing spline is returned by the solver in R, which is rounded to the nearest integer when plotting. To estimate the degree of freedom of trend filtering, for each choice of $\lambda$, we repeated the experiment for 10 times and compute the average number of nonzero knots as estimated degree of freedom. For neural networks, we use the definition (Tibshirani, 2015):

$$2\sigma^2 \mathrm{df} = \mathbb{E}\|\boldsymbol{y}' - \hat{\boldsymbol{y}}\|_2^2 - \mathbb{E}\|\boldsymbol{y} - \hat{\boldsymbol{y}}\|_2^2 \tag{31}$$

where df denotes the degree of freedom, $\sigma^2$ is the variance of the noise, $\boldsymbol{y}$ are the labels, $\hat{\boldsymbol{y}}$ are the predictions and $\boldsymbol{y}'$ are independent copy of $y$. We find that estimating (31) directly by sampling leads to large error when the degree of freedom is small. Instead, we compute

$$2\sigma^2 \hat{\mathrm{df}} = \hat{\mathbb{E}}\|\boldsymbol{y}_0 - \hat{\boldsymbol{y}}\|_2^2 - \hat{\mathbb{E}}\|\boldsymbol{y} - \hat{\boldsymbol{y}}\|_2^2 + \hat{\mathbb{E}}\|\boldsymbol{y} - \bar{y}_0\|_2^2 - \|\boldsymbol{y}_0 - \bar{y}_0\|_2^2 \tag{32}$$

where $\hat{\mathrm{df}}$ is the estimated degree of freedom, $\mathbb{E}$ denotes the empirical average (sample mean), $\boldsymbol{y}_0$ is the target function and $\bar{y}_0$ is the mean of the target function in its domain.

**Proposition 20.** *The expectation of (32) over the dataset $\mathcal{D}$ equals (31).*

*Proof.*

$$2\sigma^2\hat{\mathrm{df}} = \mathbb{E}_{\mathcal{D}}[\hat{\mathbb{E}}\|\boldsymbol{y}_0 - \hat{\boldsymbol{y}}\|_2^2 - \hat{\mathbb{E}}\|\boldsymbol{y} - \hat{\boldsymbol{y}}\|_2^2 + \hat{\mathbb{E}}\|\boldsymbol{y} - \bar{y}_0\|_2^2 - \|\boldsymbol{y}_0 - \bar{y}_0\|_2^2]$$

$$= \mathbb{E}\|\boldsymbol{y}_0 - \hat{\boldsymbol{y}}\|_2^2 - \mathbb{E}\|\boldsymbol{y} - \hat{\boldsymbol{y}}\|_2^2 + \mathbb{E}_{\mathcal{D}}[\hat{\mathbb{E}}[(\boldsymbol{y} - \boldsymbol{y}_0)(\boldsymbol{y} + \boldsymbol{y}_0 - 2\bar{y}_0)]]$$

$$= \mathbb{E}\|\boldsymbol{y}_0 - \hat{\boldsymbol{y}}\|_2^2 - \mathbb{E}\|\boldsymbol{y} - \hat{\boldsymbol{y}}\|_2^2 + \mathbb{E}\Big[\sum_{i=1}^{n}\epsilon_i(2y_i + \epsilon_i - 2\bar{y}_0)\Big]$$

$$= \mathbb{E}\|\boldsymbol{y}_0 - \hat{\boldsymbol{y}}\|_2^2 - \mathbb{E}\|\boldsymbol{y} - \hat{\boldsymbol{y}}\|_2^2 + n\sigma^2$$

$$= \mathbb{E}\|\boldsymbol{y}' - \hat{\boldsymbol{y}}\|_2^2 - \mathbb{E}\|\boldsymbol{y} - \hat{\boldsymbol{y}}\|_2^2$$

where $\mathcal{D}$ denotes the dataset. In the third line, we make use of the fact that $\mathbb{E}[\epsilon_i] = 0, \mathbb{E}[\epsilon_i^2] = \sigma^2$, and in the last line, we make use of $\mathbb{E}[\epsilon_i'] = 0, \mathbb{E}[\epsilon_i'^2] = \sigma^2$, and $\epsilon_i'$ are independent of $y_i$ and $y_{0,i}$ $\quad\square$

One can easily check that a "zero predictor" (a predictor that always predict $\bar{y}_0$, and it always predicts 0 if the target function has zero mean) always has an estimated degree of freedom of 0.

In Figure 3(h)(i), we take the minimum MSE over different choices of $\lambda$, and plot the average over 10 runs. Due to optimization issue, sometimes the neural networks are stuck at bad local minima and the empirical loss is larger than the global minimum by orders of magnitude. To deal with this problem, in Figure 3(h)(i), we manually detect these results by removing the experiments where the MSE is larger than 1.5 times the average MSE under the same setting, and remove them before computing the average.

### H.4 MORE EXPERIMENTAL RESULTS

#### H.4.1 REGULARIZATION WEIGHT VS DEGREE-OF-FREEDOM

As we explained in the previous section, the degree of freedom is the exact information-theoretic measure of the generalization gap. A Larger degree-of-freedom implies more overfitting.

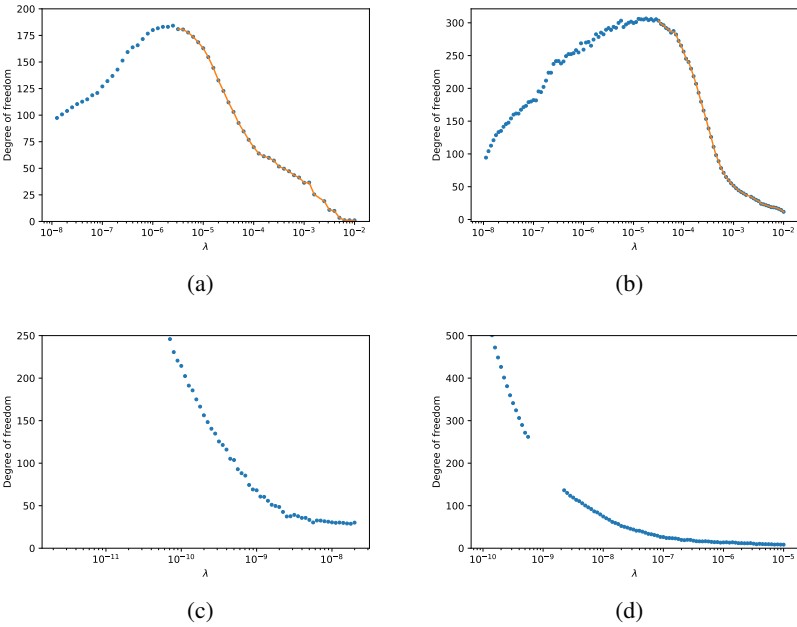

Figure 4: The relationship between degree of freedom and the scaling factor of the regularizer $\lambda$. The solid line shows the result after denoising. (a)(b)in a parallel NN. (c)(d) In trend filtering. (a)(c): the "vary" function. (b)(d) the doppler function.

In figure Figure 4, we show the relationship between the estimated degree of freedom and the scaling factor of the regularizer $\lambda$ in a parallel neural network and in trend filtering. As is shown in the

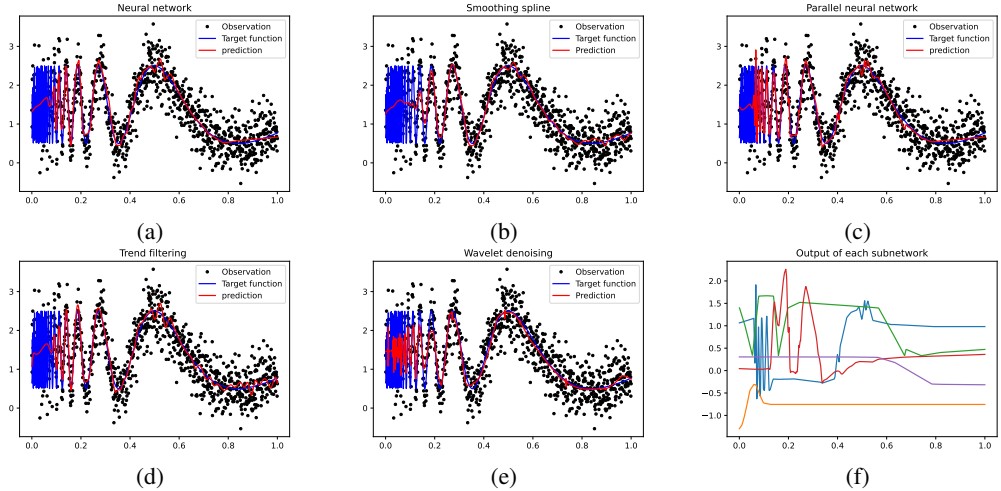

Figure 5: More experiments results of Doppler function.

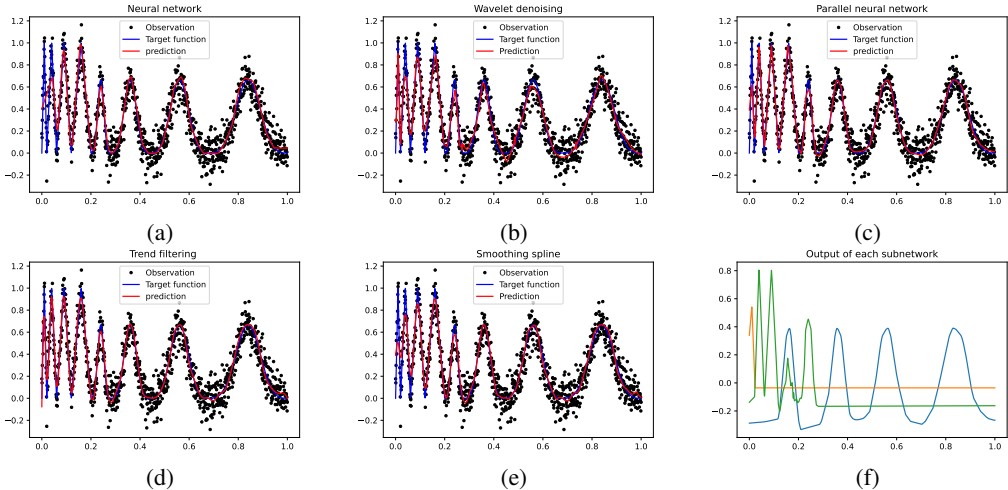

Figure 6: More experiments results of the "vary" function.

figure, generally speaking as $\lambda$ decreases towards $0$, the degree of freedom should increase too. However, for parallel neural networks, if $\lambda$ is very close to $0$, the estimated degree of freedom will not increase although the degree of freedom is much smaller than the number of parameters — actually even smaller than the number of subnetworks. Instead, it actually decreases a little. This effect has not been observed in other nonparametric regression methods, e.g. trend filtering, which overfits every noisy datapoint perfectly when $\lambda \to 0$. But for the neural networks, even if we do not regularize at all, the among of overfitting is still relatively mild $30/256$ vs $80/1000$. In our experiments using neural networks, when $\lambda$ is small, we denoise the estimated degree of freedom using isotonic regression.

We do not know the exact reason of this curious observation. Our hypothesis is that it might be related to issues with optimization, i.e., the optimizer ends up at a local minimum that generalizes better than a global minimum; or it could be connected to the "double descent" behavior of DNN (Nakkiran et al., 2021) under over-parameterization.

### H.4.2 DETAILED NUMERICAL RESULTS

In order to allow the readers to view our result in detail, we plot the numerical experiment results of each method separately in Figure 5 and Figure 6.

### H.4.3 PRACTICAL EQUIVALENCE BETWEEN THE WEIGHT-DECAYED TWO-LAYER NN AND L1-TREND FILTERING

In this section we investigate the equivalence of two-layer NN and the locally adaptive regression splines from Section B. In the special case when $m = 1$ the special regularization reduces to weight decay and the non-standard truncated power activation becomes ReLU. We compare L1 trend filtering (Kim et al., 2009) (shown to be equivalent to locally adaptive regression splines by Tibshirani (2014)) and an overparameterized version of the neural network for all regularization parameter $\lambda > 0$, i.e., a regularization path. The results are shown in Figure 7. It is clear that as the weight decay increases, it induces sparsity in the number of knots it selects similarly to L1-Trend Filtering, and the regularization path matches up nearly perfectly even though NNs are also learning knots locations.

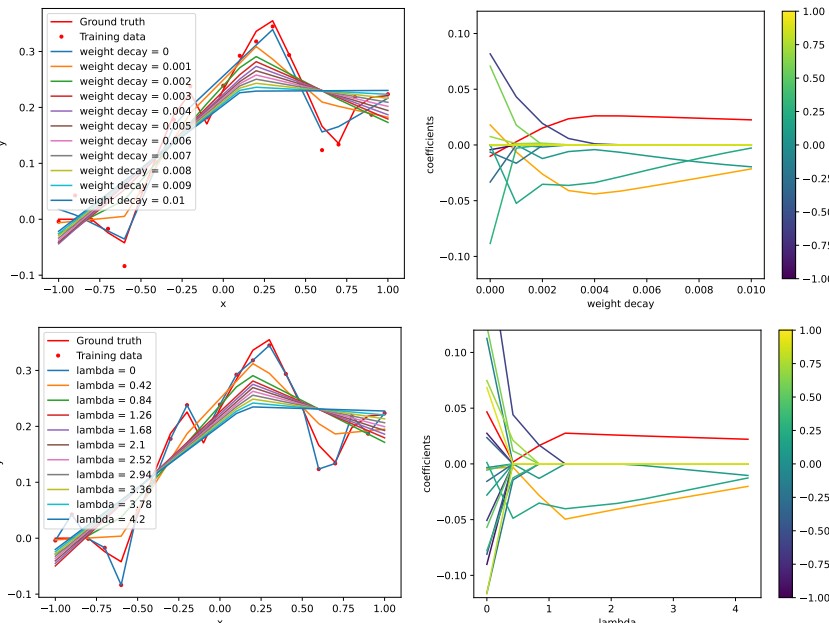

Figure 7: Comparison of the **weight decayed ReLU neural networks (Top row)** and **L1 Trend Filtering (Bottom row)** with different regularization parameters. The left column shows the fitted functions and the right column shows the *regularization path* (in the flavor of Friedman et al. (2010)) of the coefficients of the truncated power basis at individual data points (the free-knots learned by NN are snapped to the nearest input $x$ to be comparable).

