# OpenReview forum: "Deep Learning meets Nonparametric Regression: Are Weight-Decayed DNNs Locally Adaptive?"
_ICLR.cc/2023/Conference — ICLR 2023 poster_

### Official Review · Reviewer_ikZk · 2022-10-22

**Confidence:** 4
**Correctness:** 2
**Technical Novelty And Significance:** 3
**Empirical Novelty And Significance:** Not applicable
**Recommendation:** 5

**Clarity, Quality, Novelty And Reproducibility:**

This paper is well-written. Apart from the above mentioned issues, I have other questions that

1. I agree with the authors that kernel methods, e.g., NTK, lacks local adaptivity to some extent due to RKHS. I’m wondering that, the mean field regime (instead of NTK) is also suboptimal to local adaptivity?


2. For the random design, the proof of estimation error can be similarly established by [1]?

[1] Schmidt-Hieber, Johannes. "Nonparametric regression using deep neural networks with ReLU activation function." The Annals of Statistics 48, no. 4 (2020): 1875-1897.


**Strength And Weaknesses:**


**Pros:**

1. This work allows for over-parameterization without the extra sparsity condition
The obtained bound of the covering number is independent of the number of subnetworks, which is substituted by the Frobenius norm.

**Cons:**

1. The motivation on locally adaptivity makes sense, comparing linear estimators and DNNs. Nevertheless, the motivation on the comparison of (Suzuki ICLR2019)  is relatively weak. The authors claim that Suzuki’s work requires certain conditions on the width, the depth, the sparsity to achieve the minimax rate on deep ReLU networks, which is different from practical uses. Nevertheless, this work does not focus on this issue but centers around another architecture, i.e., parallel neural networks. Even though parallel neural networks have theoretical and empirical applications as the authors suggested, the “correct” motivation is to solve certain conditions in Suzuki’s work under the same architecture.

More importantly, this work also requires several conditions on the width, the depth, etc. In the “No architecture tuning” part, the required architecture of the model still depends on $\alpha$, see Theorem 1. It is hard for me to accept the authors’ motivation.


2. In theorem 1, the number of subnetworks is exponentially large enough w.r.t the input dimension $d$. This is unacceptable in both theory and practice.


3. In Proposition 4, the equivalence between Eq. (2) and Eq. (12) is imprecise via the Lagrangian’s method in optimization. The constraint value P doesn't appear in Eq. (2) and it is also not clear how the dual variable is handled. In fact P (as well as P’) is unknown or uncomputable, which also makes the upper bound of the covering number appear infeasible in Theorem 5 depending on P’. It is ok for ease of description but is not rigorous in a theory paper corresponding to the first contribution.

4. After reading this paper, I think the technical contribution is to derive the covering number bound of the parallel neural networks for the estimation error, i.e., Theorem 5 and Lemma 6. The proof for approximation error follows Yarotsky (2017); Suzuki (2018). However, this paper works under a fixed design, which is much easier to derive the estimation error.


**Summary Of The Paper:**

This paper extends Suzuki’s work (ICLR2018) to parallel neural networks in the Besov space as well as BV functions, studies the generalization properties in terms of approximation error via B-spline basis and estimation error via covering number, and demonstrates the $\ell_2$ norm (weight decay) in parallel ReLU neural networks is equivalent to a certain sparse regularization term.


**Summary Of The Review:**

This work has some good results on the sparsity, the separation between linear estimators and DNNs in Besov spaces as well as BV functions. Nevertheless, the derived results appear vacuous in terms of imprecise Lagrangian dual and curse of dimensionality of the input dimension.

---

> ### Author Response · Authors · 2022-11-10
> **Respond to the reviewer**
>
> We would like to thank the reviewer for the constructive comment. As for the comments:
>
> 1.
> This work focuses on the same topic as Suzuki's work, and we borrow some techniques in proof, i.e. constructing a neural network to approximate functions in Besov space.
> Our motivation is to eliminate the need to tune sparsity constraint to achieve the minimax rate, and replace it with a fixed sparsity pattern (block-sparseness, with is equivalent to the parallel NN) together with the standard regularization.
> Although it was not explicited stated, the model required to approximate a target function in Suzuki's work is actually a parallel neural network.
> We believe our method is more practical, since enforcing the sparsity constraint during training is quite difficult.
>
> Since the discussion in ``no architecture tuning'' part leads to confusion, we have modified the discussion in that part.
> Basically, with some mild constraint on the smoothness parameter ($\alpha$ is not larger than an estimation), we show that parallel NNs can adapt to Besov class with any smoothness parameter $\alpha$.
> Notably, one do not need to tune the number of subnetworks $M$. The only requirement is that $M$ is large enough.
> As is shown in the paper, the model after training will be sparse in terms of the number of "active'' subnetworks, so further increasing $M$ will not incur any penalty in MSE.
> This is the most important difference, as in Suzuki's work, the estimation error depends on the number of non-zero elements, so it requires explicitly-enforces sparsity, while our work only requires implicitly-enforced sparsity induced by $\ell_2$ regularization.
>
>
> 2.
> In response to this comment, we have slightly changed our way to construct the parallel NN, such that $M$ does not increase exponentially with $d$, while the width of each subnetwork  $w$ have changed to be fixed and larger than $O(md)$.
> Yet we would like to note that in our setting $d$ should not be very large.
> This is because any estimator in Besov space requires exponential large number of samples to achieve desired accuracy.
> Should $d$ be large, stronger assumptions are required, eg. the data is low rank.
>
> 3.
> Lagrangian's method allows converting a constrained problem to a regularized problem or vice versa, so it draws a one-to-one mapping between $\lambda$ in (2) with $P$ in (12) ((13) in the rebuttal revision).
> See https://stats.stackexchange.com/questions/134763/what-is-the-connection-between-regularization-and-the-method-of-lagrange-multipl
> Actually,
> $P$ can be computed as in theorem 8. We admit that $\lambda$ cannot be computed analytically, but it can be empirically determined, eg. using cross validation.
>
> 4. In this paper, we mainly focus on the fixed design problem which is standard in non-parameter regression problems. We can easily extend this result to random design and we added it as Theorem 17 (in the appendix) in the rebuttal revision.
>
> Other questions:
> 1. Unfortunately we cannot provide an answer to this question. Neither our result nor theorem of linear estimators  extend to the mean field regime of NNs, and there is no existing work on that to the best of knowledge. We believe this question worth studying in the future.
>
> 2. Please see our respond to comment 4.

---

> > ### Comment · Reviewer_ikZk · 2022-11-10
> > **unsolved issues**
> >
> > Thanks for the authors’ feedback.
> >
> > - I agree with the authors’ current motivation to eliminate the need for the sparsity constraint, but in this case the Lagragian’s method is very important in a theory paper.
> >
> > - For Lagrangian dual, the relationship between the original problem with constraints and the dual problem can be found in [Boyd. Convex Optimization; Chapter 5]. I need mention two issues:
> >
> > *1)* the optimal solution and the Lagrangian multiplier (i.e., the regularization parameter $\lambda$) requires a min-max optimization rather than assuming that there exists a suitable $\lambda$, and then separately solve a minimization optimization problem.
> >
> > *2)* Neural networks training is often a non-convex optimization problem, only the weak duality holds. In this case, these two problems are not equivalent. I’m not sure this can be approximately equivalent via some hyper-parameter cross validation.
> >
> > - Regarding the exponential order of d, I appreciate the authors’ effort on the improvement to a polynomial order. When checking Proposition 4, the output layer $a_j$ has a bounded $2/L$ norm but depends on a unknown $P’$ from the “so-called” Lagrangian dual method. This makes me further devalue this work as avoiding the sparsity constraint when compared to Taiji’s work is a main point in my view.
> >
> > Based on the above points, I still maintain my evaluation. It is unclear to me why the authors don’t directly focus on the constrained optimization of neural networks training in Eq. (2). I understand that this is a bit far from the commonly used regularization-based scheme, e.g., weight decay. But the authors assumed that the global minimum can be directly achieved, so I don’t find some significant justification to focus on weight-decay and then change it to a constraint optimization in a non-rigorous dual equivalence. Actually it makes sense to focus on a norm-constrained style from the perspective of deep learning theory in approximation and generalization, e..g., norm-based capacity of DNNs with path norm, spectral norm.
> >
> > Besides, I appreciate authors who honestly mention the unclear relationship between current method and mean-field regime. But I think the authors could explain the current initialization scheme and training scheme lead to lazy-training or feature learning. This issue deserves some discussion in this paper when compared to the commonly-used NTK and the mean field regime in DL theory.

---

> > > ### Author Response · Authors · 2022-11-15
> > > **Respond to the comment:**
> > >
> > > We would like to thank the reviwer for pointing out our misunderstanding about Lagrange method. We have modified our proof, and provide a mild assumption, which is a sufficient condition that the regularized problem achieves the claimed error rate.
> > > We would like to make the following notes:
> > >
> > > (1) Our results are valid without additional assumptions for the constrained version of the problem. By tuning only the L2-constraint on the weight, we can obtain near-optimal locally-adaptive rates for many function classes.  This itself is novel and shows that there is something new deep learning brings to a classical problem.
> > >
> > > (2) For the same result to work for the regularized version of the problem rigorously, we need an additional assumption  (See Lemma 17 in the updated submission).  The assumption is implied by a condition that the regularizer term does not jump too much as a function of $\lambda$, or the generalization gap is not too large.
> > >
> > > (3)  We believe this issue with constrained / regularized versions of the algorithm is more prevalent and not specific to this work. We think what we looked into might be useful in establish some weak conditions and more generically applicable to such problems. It might be worthwhile of a dedicated research article.
> > >
> > > (4) As is discussed in the paper, our method shows that training a parallel NN is equivalent to a sparse dictionary learning problem where the dictionary is learnable.
> > > This idea is similar to feature learning.
> > > However, we still need to mention that our analysis does not cover the training dynamics, so it is not directly comparable to the NTK or the mean field regime in DL theory.

---

> > > > ### Comment · Reviewer_ikZk · 2022-11-15
> > > > **increase my score to 5**
> > > >
> > > > Thanks for the authors' response on the regularized/constrained optimization.
> > > >
> > > > To me, the most important message from this work is that, the weight-decay regularization in DNNs leads to sparsity.
> > > > I increase my score to 5 (actually I remained ambivalent about giving 6 because of this message.)
> > > >
> > > > btw, there is a missing cross-referencing in Step 3 at page 23.

---

> > > > > ### Author Response · Authors · 2022-11-15
> > > > > **Thank you! Could you be a champion?**
> > > > >
> > > > > Thank you for increasing the score and for highlighting one of the important take-home messages of this paper: “Weight decay induces sparsity!”  This is really the technical reason why we got adaptivity and an error bound that doesn't depend on how overparameterized the model is, though formalizing these results is nontrivial (thus our paper).
> > > > >
> > > > > > Re: "(actually I remained ambivalent about giving 6 because of this message.)"
> > > > >
> > > > > If we understand your comments correctly, you were willing to increase to 6 now that the technical aspects are cleared up, but are ambivalent because you find the main message above is now weakened?
> > > > >
> > > > > If so, we would like to further respond to this by clarifying the following points:
> > > > >
> > > > > (1) The main message you spoke of — our claim that weight decay regularization in PNN is equivalent to an L_p sparsity *regularized* linear regression with representation learning — does *not* require any additional assumptions, thus not actually weakened.  It is proven in Proposition 4.
> > > > >
> > > > > (2) The additional assumption (from lemma 17) is only required for technical reasons when proving the error bounds for the regularized objective via our results for the constrained objective, because of the issue you pointed out.
> > > > >
> > > > > (3) The constrained version of the objective function is equally popular. For example, the original Lasso paper by Tibshirani (1996) studied the L1-constrained version of the problem. The regularized version was only popular much later. For our problem, an L2 constraint that induces implicit sparsity is still somewhat surprising, don’t you think so?
> > > > >
> > > > > (4) We believe our inability to directly analyze the regularized version of our problem without this additional assumption is due to the limitation of available theoretical tools as of today, thus only an artifact of the analysis.  Our experiments clearly suggest that the regularized objective induces very sparse support, and achieves near-optimal MSE.
> > > > >
> > > > > We hope the above resolves your remaining concern and you could kindly consider supporting the paper more strongly!
> > > > >
> > > > > Regardless of the outcome, we sincerely thank you for your insightful comments that led us to a significant improvement in our manuscript!

---

### Official Review · Reviewer_wKTh · 2022-10-25

**Confidence:** 4
**Correctness:** 4
**Technical Novelty And Significance:** 3
**Empirical Novelty And Significance:** 3
**Recommendation:** 6

**Clarity, Quality, Novelty And Reproducibility:**

Some of the technical contributions need highlighting. For example, the conversion of weight decay objective to the constraint form in Equation (5) can be highlighted as a contribution -- if it does not appear in Parhi & Nowak (2021a). Another example is the decomposition of \mathcal{F}_{\parallel} and \mathcal{F}_{\perp}.

A discussion on why parallel neural networks can only achieve the optimal rate in the asymptotic sense ($L \to \infty$) is helpful. Is it because the weight decay introduces additional bias?

In Theorem 1, it is good to state how to choose weight decay $\lambda$.

Lemma 6 is relative difficult to understand, mainly because the first sentence begins with an equation.

Overall, the paper is well written. There are some typos and typeset issues, for example, "errorand" --> "error and" above Section 4.2

**Strength And Weaknesses:**

============= Strength =============

The paper is clearly written with both theoretical and empirical results. The theoretical results are correct and empirical results are supportive.

The adaptability of neural networks is studied in many recent works, including its adaptability to data geometry and adaptability to structures in function spaces. This paper considers parallel neural networks and uses weight decay for controlling the complexity of the network, which circumvents sparsity constraints in many existing network approximation results.

============= Weakness =============

There is no related work section. Some of papers share similar ideas to parallel neural networks, for example, "Approximation and Non-parametric Estimation of ResNet-type Convolutional Neural Networks" and "Besov Function Approximation and Binary Classification on Low-Dimensional Manifolds Using Convolutional Residual Networks" both rely on implementing block sparse feedforward networks. The resulting (residual) convolutional network does not need a sparsity constraint in training.

The obtained rate of estimation is slightly slower than the optimal rate, which is attainable (up to log factors) in many recent works on using ReLU neural networks for nonparametric regression.

From experiments, parallel neural networks seem to achieve comparable performance to neural networks and conventional methods.


**Summary Of The Paper:**

The paper analyzes parallel neural networks for nonparametric regression (including approximation error and regression error analysis). Suppose the regression function has nonhomogeneous regularities, e.g., Besov functions or functions of bounded variation. Parallel neural networks trained with weight decay achieves near optimal estimation, indicating the adaptability of parallel neural networks.

**Summary Of The Review:**

I think the paper is interesting. I am currently giving a slightly negative rating, due to some concerns of the results and missing references. I am happy to raise to a positive rating after paper revision and an effective author feedback.

---

> ### Author Response · Authors · 2022-11-10
> **Respond to the reviewer**
>
> We would like to thank the reviewer for the suggestions constructive comment. Here is the clarification about the comments:
>
> 1. Due to page limit, we placed the related work section in the appendix and we added a pointer to that in the introduction section in the rebuttal revision. As is suggested by the reviewers, we have also added a table comparing our result with previous work.
> We also cited the works suggested by the reviewers in the rebuttal revision, and compared these works with ours.
> Although these works does not require sparsity, they require the architecture of the model, including the number of (nonzero) parameters, to be tuned based on the target function and number of samples to be tuned, which is difficult in practice.
> Our work only requires tuning the regularization parameter, which we believe is easier to train in practice.
> Besides, the models required in Resnet-style works may be too deep to train in practice, Besides, the models required in Resnet-style works may be too deep to train in practice, especially when $n$ is large.
>
>
> 2. The purpose of our experiment is to demonstrate that parallel NNs achieve similar result to conventional methods. Because the conventional methods are known to be optimal in BV class, no method can outperform the conventional methods (up to a constant factor). Since neural networks have stronger adaptivity, we recognize there is a "price" to pay (higher error) and we are pleased to find that the "price" is not too large. Our result does not cover standard neural networks and there is not existing theorem predicting the performance of that to the best of our knowledge.
>
> 3. The reason for gap between the estimation error and the minimax rate is that the minimax rate can be achieved by an $\ell_0$ sparse model, while the parallel NN is equivalent to an $\ell_p$ sparse model, which is an approximation to $\ell_0$.
> As $L$ increases, $p$ gets closer to 0, leading to lower gap in error rate.
>
>
>
> We have modified the paper based on the reviewer's suggestion. Specifically, we have also highlighted our technical contributions in the introduction part, added discussion why there is the gap between error rate of NN and the minimax rate, added as section (lemma 17) which discusses how to choose $\lambda$, and corrected the expression in Lemma 6.

---

### Official Review · Reviewer_YNWp · 2022-10-25

**Confidence:** 2
**Clarity, Quality, Novelty And Reproducibility:** The paper is mostly clear, novel, and…
**Correctness:** 3
**Technical Novelty And Significance:** 3
**Empirical Novelty And Significance:** 3
**Recommendation:** 6

**Strength And Weaknesses:**

Strength:
  1. The paper is well-written and mostly clear.
  2. The theoretical analysis provided by the authors are, as far as I am aware of, novel and interesting.
  3. The obtained results do provide some insights on, for example, why deep networks do no overfit and why deep networks perform better than shallow networks.

Weaknesses:

 First off, I want to say that I am very new to the area of non-parametric regression and the function spaces the authors use in this paper. I have some experience with deep learning theory but this direction is completely new to me. Therefore, I might not be the best reviewer for this paper. Some of my comments might be unfair/wrong and I hope the authors and the other reviewers can help me understand this paper a little more during the discussion period.

  Now back to this paper, I do have some questions about the setting and the results the authors provide.

  1. The authors have chosen to optimize a set of parallel feedforward networks with $L_2$ regularization (I don't really like using the word "weight decay" here because there is no mentioning of the optimizer, e.g. SGD/Adam, and thus weight decay does not really exist). I wonder why the authors didn't include the regularization for the bias in Eqn. (2)? After looking at the Appendix, I think the authors do analyze the network with the bias term, so why is the regularization for bias missing? I think in practice, people have both regularization for weights and bias.

  2. In Theorem 1, I am confused by the statement "with proper choice of the parameter of weight decay $\lambda$". How large does the parameter need to be? Does it need to be time-varying, for example? If the choice of $\lambda$ needs to depend on the other parameters such as $M, L, p, q$, please state it explicitly in the theorems so that we can understand it more clearly.

  3. I am a little confused by the results in Corollary 2 and 3. For example, in Cor 2, although the $o(1)$ term is negligible, does it mean that the rate of deep neural network is worse than that of traditional nonparametric regression models? Are there models that can achieve the minimax rates that the authors have state Page 3? I think it might be better to have a table to compare the results in this paper with the results in the literature. Also, what is going to happen if $L > 100 C \log n$? For very deep neural networks, do the theorems still hold?

  4. For Cor 3, although I know that most of the paper is focused on $L\gg 2$, the authors might want to mention that the results here fail for $L=1, 2$. Also, the claim that "DNN has an advantage over kernels" is a little confusing, shouldn't there be an assumption on $L$ (L> some function of m) so that the bound for DNN is strictly better than that of kernels? I hope the authors can state it more clearly.

  I do not have enough time to go through all the proof details, but the proof overview section does give me enough intuition of how the theorems are derived. The experimental results are a good supplement to the theory.

**Summary Of The Paper:**

This paper proposes to analyze the theory of deep neural network from the perspective of non-parametric regression. The authors show that training an $L$ layer neural network with weight decay is basically the same as penalized regression. And in the case of functions in the Besov space and bounded variation space, neural networks achieves almost near-optimal rates. When the number of layers increase, the error decreases and thus explains why deep networks performs better than shallow ones. The authors also validate some of their claims using experiments.

**Summary Of The Review:**

Overall, this seems to be a good paper with nice contributions. However, I am confused by some settings and theoretical results of the authors. Again, I am no expert in this field, and if there is anything wrong in my review, please correct me and I am happy to increase my score.

---

> ### Author Response · Authors · 2022-11-10
> **Respond to the reviewer**
>
> We would like to thank the reviewer for the constructive comment. As for the questions:
>
> 1. Empirical, the performance difference between regularizing the bias and not is often insignificant.
> There are also many works that leave the bias term not regularized, for example, when training BERT.
> Technically,
> We leave the bias term not regularized so that the expression of the equivalent model (proposition 4) is cleaner.
> This also helps us make a fair comparison with other non-parametric regression methods, as most of those methods have a similar null space (function space that is not regularized).
> On the other hand, if we regularize the bias instead, we can get a similar result as long as the target function satisfy a mild additional constraint (eg. having a bounded image), as neural networks will be biased in predicting the constant term $f(x)=c$ in the target function.
> From the technical perspective, handling these unregularized biases is one of our key technical contributions because they are not constrained and do not have a bounded covering number.
> We have added a short paragragh to talk about that in the rebuttal revision.
>
>
> 2.
> We provide an explicit choice of $\lambda$ in Lemma 17 in the rebuttal revision.
> On the other hand, it can be determined empirically, e.g. using cross validation. We have added a discussion about it in the rebuttal revision ("Hyperparameter tuning'' part).
>
> 3. Our result does indicate that the rate of deep neural network is slightly slower than that of traditional nonparametric regression models, which achieves the exact minimax rate. Yet deep neural networks still have faster rate than any linear method, eg. kernel method.
> There are some works that show that a sparse neural network can achieve the minimax rate up to a logarithm factor, for example, Suzuki ICLR2019,
> but the methods to realize the error rate are different.
> Previous methods requires tuning the number of parameters so as to achieve the desired error rate, while our method allows an overparametered model and requires tuning only the regularization factor.
> We believe our method (tuning the regularization factor) is easier to implement than the other methods (tuning the model architecture).
>
> Theorem 1 holds for any $L\geq 3$.
> When $L$ is very large (eg. $L > 100C\log n$), the theorem will still hold, yet the estimation error will dominant the MSE so the error will increase with $L$ (so-called overfitting).
>
> We have added a table to compare our result with other works.
>
> 4. When $L=1$, the model reduces to a linear model. When $L=2$, the model is a standard two-layer neural network, which is discussed in Parhi \& Nowak 2021a,c, and also Section B in the appendix.
>
> We have added a discussion showing the condition that the NNs have strictly lower accuracy than any linear method.
> When $L > O(m^2)$, the error of NNs decreases with $n$ faster than that of the linear method. In this case, if $n$ is large enough, there exist $L$ such that the error of NN is lower than that of linear method. Note that the second term of MSE of NN decreases exponentially with $n$, so it is neglectable when $L \geq O(\log n)$, and the first term increases only polynomially with $L$.

---

### Official Review · Reviewer_UB7m · 2022-10-27

**Confidence:** 3
**Correctness:** 3
**Technical Novelty And Significance:** 3
**Empirical Novelty And Significance:** 2
**Recommendation:** 6

**Clarity, Quality, Novelty And Reproducibility:**

I can mostly follow the paper but was confused by some details from time to time.

**Strength And Weaknesses:**

Strength: The authors can show that a specific kind of neural net achieves minimax rate for an interesting function family. Some of the techniques appear to be significant (e.g., hacking into covering numbers).

Weakness: there is room for improvement in presentation.

**Summary Of The Paper:**

This paper examines training a specific neural network (namely parallel neural net) using weight decay and shows that the error rate matches the minimax bound for a specific class of functions (i.e., functions with bounded variation).

It looks like a quite interesting effort and the result also appears to be significant. But I worry the paper is not presented at the level for general audience in ICLR. For example, I have a few questions/confusions after reading the submission:

1. What exactly is weight decay? I actually could not find its definition/elaboration except for eq. 2, which is just L2 regularization? I was guessing it could relate to learning rate, momentum, some dynamic shrinkage on the fly...
2. Do we assume that Eq.2 can be optimally solved? It is a minimax result related to sample complexity so I believe the authors assume that Eq. can be optimally solved --- I think sometimes ignoring computational tractability is fine (and perhaps had been a standard practice in statistics) but it would be helpful to bring this upfront.
3. What's so special about parallel NN? I think most audience wants to understand at a higher level where the magic happens and why the magic wont happen for the standard MLP, e.g., only limit of existing theoretical tools or parallel NN possesses some special structure?
----
Thank you for addressing the presentation concern. I have raised my scores to 6 -- do not feel am an expert in this area so no confidence to move to 7.





**Summary Of The Review:**

I think the substance is above the bar of ICLR; the revision addressed my presentation concern.

---

> ### Author Response · Authors · 2022-11-10
> **Respond to the reviewer**
>
> We would like to thank the reviewer for the constructive comment. As for the questions:
>
> 1. Weight decay is equivalent to adding L2 regularization during training. In order to reduce confusion, we have included the definition and replaced the ``weight decay`` with ``L2 regularization`` in the majority of the content in the rebuttal revision.
>
> 2. We ignore the computation issue and focus on the optimal solution in eq. 2. In practice, it is not guaranteed that the optimal solution can be found, but it has been found that when training a neural network with gradient descent, with high probability the solution is close to the global optimal solution.
> We have added a section discussing about that in the rebuttal revision.
>
> 3. The most important property of parallel NNs we make use of is that in parallel NN, AM-GM inequality can be applied to each subnetwork. Using this property, we prove that a parallel NN trained with standard L2 regularization is equivalent to an $\ell_p$ sparse model (proposition 4) due to its special structure. In a standard MLP, the AM-GM inequality can only be applied to the entire NN, so it does not lead to sparsity, and our analysis does not extend to MLPs. Existing theorem neither prove nor deprove that standard MLPs have the same property, so we do not know if MLPs without sparsity constraint are locally adaptive or not.
> This is an interesting and important question that deserves future study.

---

### Author Response · Authors · 2022-11-10
**General response**

We would like to thanks the reviewers for the suggestions and comments. Here are some general response to the comments:

1. We do not consider computation and focus on statistical theory of the argmin. We note that the implicit regularization of SGD behaves like weight decay (or square L2 regularization), thus even though we focus on the exact minimizer of the regularized objective it may explain the behavior of SGD in practice. In our experiments, we find that when the number of subnetworks is proportional to n, we can consistently find good solutions with standard deep learning optimizers and that our theory is supported by the experiments.

2. Our results are new for this problem, but we are not the first to claim the separation of NN from kernels nor separation of deep NN from shallow ones. We discuss the differences briefly to avoid any confusion.

* Separation of NN from kernels. It was known that NN can be more sample-efficient than kernel methods, e.g., https://arxiv.org/abs/1905.10337 but the construction focuses on a particular set of example that require input dimension d to be large and output dimension k>= 2. Our construction, however, is simpler and works with univariate input and univariate output, and works even in the fixed design case with gaussian noise.

* It is well-known that deeper NN can approximate functions that shallow NNs cannot: https://mjt.cs.illinois.edu/dlt/#separating-shallow-and-deep-networks Our results complement the existing work but showing that for Besov class, deeper NN has the advantage of adaptivity in that we can just use ReLU and the standard weight decay (instead of tuning the number of parameters in the model) and *simultaneous* recover the optimal rates for many nonparametric classes.

---

> ### Author Response · Authors · 2022-11-15
> **The main contribution**
>
> Regarding the contribution of our paper compared to previous works on this topic:
>
> (1) We prove a unique property of parallel NN: we can apply AM-GM inequality on each subnetwork.
> This is different from standard neural networks where one can only apply AM-GM inequality on the entire model, and is the key technique to prove its equivalence to the $\ell_p$ sparse model.
>
> (2) We provide a bound on the estimation error via covering number (metric entropy) that does not depend on the number of (nonzero) parameters in a neural network, but instead on the norm of the parameters.
> This bound is the key to our proof, and we believe this method can find a more general application in the theory of deep learning and can of separate interest.
>
> (3) We provide an explicit choice of $\lambda$ in Lemma 17 in the rebuttal revision.
> We prove that under a mild condition, for any constrained optimization problem, with this choice of $\lambda$, the MSE of the solution to the regularized optimization problem not larger than that of the constrained optimization problem up to a factor of constant.
> In contract to other works that focus on constrained problems, this technique can be used to analysis the regularized problems directly which are easier to solve empirically, and should have a general application.

---

### Decision · Program_Chairs · 2023-01-20

**Decision:**

Accept: poster

**Justification For Why Not Higher Score:**

As I wrote in the summary section, this paper's novelty is rather limited. The parallel structure has been implicitly exploited by Yarotsky (2017), Suzuki (2018) and Schmidt-Hieber (2020). There are some theoretical ambiguity in the connection between the regularization formulation and constraint formulation.

**Justification For Why Not Lower Score:**

This paper's main message is novel. The automatic model tuning and its statistical optimality is an important topic in the literature. The theoretical finding is well justified by the numerical experiments.
The original submission had some minor issues as pointed out by the reviewers, but they were mostly addressed during the rebuttal. Then, the paper's quality became sufficiently good for acceptance.

**Metareview: Summary, Strengths And Weaknesses:**

This paper gives a new nonparametric estimation error bound of deep learning where the true function is included in the Besov or BV space. This paper considers a model of a parallel NN and shows that the L2-norm regularized empirical risk minimizer achieves the optimal rate without sparsity constraint that has been imposed by several existing work.

**Strength:** This paper explicitly shows the near minimax optimal rate without sparsity constraint when $L^2$ regularization is imposed. This is because the layer-wise $L^2$ regularization yields a sparse regularization on the coefficient of each block sub-network. In other words, the network size has mild dependency on the smoothness parameter and the optimal structure is automatically obtained without cross validation. This kind of automatic model selection ability has not been exploited in the existing literature. It somehow justifies why deep learning rarely overfit in practice. The theoretical characterization is well justified by numerical experiments.

**Weakness:** The technical tools heavily depends on previous work such as Yarotsky (2017), Suzuki (2018) and Schmidt-Hieber (2020). Indeed, their proof essentially utilizes the parallel NN structure and does not require sparsity constraint (see also [1]). I also would like to point out that a non-sparse fully connected network structure is also investigated by [2]. It would be better that a comparison to this study is included.
Another concern is that the regularization formulation and constraint formulation are not justified in a completely rigorous way.

[1] Bolcskei, H., Grohs, P., Kutyniok, G., and Petersen, P. Optimal approximation with sparsely connected deep neural networks. SIAM Journal on Mathematics of Data Science, 1(1):8–45, 2019.
[2] Jianfeng Lu, Zuowei Shen, Haizhao Yang, and Shijun Zhang: Deep Network Approximation for Smooth Functions. SIAM Journal on Mathematical Analysis, vol 53, number 5, pp.5465--5506, 2021.

Although this paper has weakness as described above, this paper's main focus is in a different point, that is, the automatic sparse model selection which is a novel view point. The concern about the relation between penalty and constraint is addressed during the rebuttal period. Although it still requires some technical assumptions, the limitation of the analysis is well discussed by referring some experimental observation.

In summary, this paper's novelty and significance are more valuable than the drawbacks. This paper's contribution is informative for the community. Hence, I recommend acceptance.

**Note From Pc:**

if the above contains the word "oral" or "spotlight" please see: "oral" presentation means -> notable-top-5% and "spotlight" means -> notable-top-25%. As stated in our emails, we are disassociating presentation type from AC recommendations